# Stabilizing black-box model selection with the inflated argmax

**Melissa Adrian**                                                    *maadrian@uchicago.edu*
*Data Science Institute, University of Chicago*

**Jake A. Soloff**                                                        *soloff@umich.edu*
*Department of Statistics, University of Michigan*

**Rebecca Willett**                                                      *willett@uchicago.edu*
*Department of Statistics, University of Chicago*
*Department of Computer Science, University of Chicago*
*NSF-Simons National Institute for Theory and Mathematics in Biology*

**Reviewed on OpenReview:** *https: // openreview. net/ forum? id=DSDWHsQLgA*

## Abstract

Model selection is the process of choosing from a class of candidate models given data. For instance, methods such as the LASSO and sparse identification of nonlinear dynamics (SINDy) formulate model selection as finding a sparse solution to a linear system of equations determined by training data. However, absent strong assumptions, such methods are highly unstable: if a single data point is removed from the training set, a different model may be selected. In this paper, we present a new approach to stabilizing model selection with theoretical stability guarantees that leverages a combination of bagging and an "inflated" argmax operation. Our method selects a small collection of models that all fit the data, and it is stable in that, with high probability, the removal of any training point will result in a collection of selected models that overlaps with the original collection. We illustrate this method in (a) a simulation in which strongly correlated covariates make standard LASSO model selection highly unstable, (b) a Lotka–Volterra model selection problem focused on identifying how competition in an ecosystem influences species' abundances, (c) a graph subset selection problem using cell-signaling data from proteomics, and (d) unsupervised $\kappa$-means clustering. In these settings, the proposed method yields stable, compact, and accurate collections of selected models, outperforming a variety of benchmarks.

## 1 Introduction

Model selection is typically formulated as a procedure to identify the model that "best" represents data from among a set of candidate models. For instance, scientists may choose among many interpretable models to identify salient variables, interactions, or policies, with the goal of developing a fundamental understanding of the natural or social phenomenon underlying experimental data. Examples of model selection include variable selection (choosing a subset of covariates from a larger pool that best explains the response variable or label), selecting a decision tree, selecting the order or memory of an autoregressive model, selecting a kernel function for a support vector machine, selecting the number of clusters in $\kappa$-means, selecting the number of factors in factor analysis, and more. The issue of model selection is a common problem in a variety of domains, including bioinformatics (Saeys et al., 2007), environmental studies (Effrosynidis and Arampatzis, 2021), and psychology (Vrieze, 2012).

In this paper, we argue that constraining a procedure to return a *single* model does not adequately express any uncertainties in the model selection process, and therefore, returning a *set* of selected model(s) is more appropriate than only ever returning one. An ideal method would return a single model as often as possible while guaranteeing some degree of *stability*, i.e., the selection should not be too sensitive to small changes in the data (Yu, 2013).

Especially in model selection scenarios where the goal is to interpret the model to learn something fundamental about the studied process, trustworthiness in the results is paramount. A necessary condition for trustworthy model selection procedures is stability. Conventional methods for model selection provide *ad hoc* procedures that do not provide theoretical stability guarantees without strong assumptions on the data or model. Our goal is to provide a method that is adaptive to the underlying uncertainty of the model selection process without placing strong assumptions on the data distribution or the models themselves.

In this paper, we propose stabilizing model selection, leveraging ideas from stabilized classification (Soloff et al., 2024b), which exhibits desirable theoretical guarantees. We provide comparisons of our proposed approach to widely used conventional model selection methods as well as explore the trade-offs of stability, accuracy, and interpretability of the model selection procedures.

## 1.1 Our contributions

In this work, we allow model selection procedures to return a *set* of candidate models, conveying uncertainty about which model is best while returning a single model as often as possible. We formulate a notion of model selection stability based on the idea that our claims about which model is best should remain logically consistent when we drop a small amount of data.

We develop a framework for stabilizing *black-box* model selection procedures by combining bagged model selection with the *inflated argmax* (Soloff et al., 2024b), which was formulated in the context of multiclass classification. **The key insight of this paper is that if we formulate model selection as a multiclass classification problem, where each class corresponds to one of the candidate models, then we can directly use the inflated argmax and inherit its corresponding theoretical stability guarantees for model selection, which are agnostic to the model class and data distribution.** Additionally, the stability guarantee holds even if the "true" model is not considered in the class of candidate models. We emphasize that our approach is highly generic and can be applied to many base model selection tasks, such as variable selection, constructing decision trees, selecting the order of an autoregressive model, and more.

We show empirically that the inflated argmax can improve black-box model selection stability with examples in clustering, decision trees, and variants of variable selection tasks. For variable selection (a special case of model selection), we show this result for data with correlated covariates, a setting known to yield instabilities in standard model selection algorithms. We compare against and outperform numerous conventional model selection approaches, described in §2.2. These conventional methods, unlike our approach, are not adaptive to the underlying level of uncertainty of the model selection procedure.

## 2 Model selection

More formally, model selection focuses on selecting one or more models from among a set of candidate models, denoted $M^+$. A *model selection procedure* is a map $\mathcal{M}$ from data $\mathcal{D}$ to a set of selected models $\hat{M} \subseteq M^+$.[1] The output $\hat{M} = \mathcal{M}(\mathcal{D})$ is a nonempty subset of $M^+$. If $\mathcal{M}$ always outputs a singleton (i.e., $|\hat{M}| = 1$), $\mathcal{M}$ is called *simple*.

For example, in variable selection (a special case of model selection), the data consists of $n$ variable-response pairs $\mathcal{D} = \{(x_i, y_i)\}_{i=1}^n$, where $x_i \in \mathbb{R}^d$. We seek to identify relevant variables (e.g., variables of $x$ most predictive of $y$.), so the model class $M^+ = 2^{[d]}$ is the power set of $[d] = \{1, \ldots, d\}$. However, in more general settings, the set of all candidate models $M^+$ can contain any class of models. For instance, in graphical model selection, $M^+$ is the set of all graphs $G$ on $d$ vertices, where an edge $(j, k)$ in $G$ encodes conditional dependence between variables $j$ and $k$ (Drton and Perlman, 2004; Friedman et al., 2008). In dynamical system identification, $M^+$ is a set of possible differential equations governing the dynamics underlying noisy data (Brunton et al., 2016). In econometrics, $M^+$ could be a collection of competing time series forecasts (Masini et al., 2023). To keep our discussion general, the model class is an abstract, countable set $M^+$.

---

[1]Note that this is a general formulation of the model selection problem, and does not assume that there is a true data generating model in $M^+$.

---

**Algorithm 1** Bagged model selection (Breiman, 1996a;b)

---

**input** Model weighting algorithm $\mathcal{A}$; data $\mathcal{D}$; ensemble size $B$; bag size $K$
  **for** $b = 1, \ldots, B$ **do**
    Construct $\mathcal{D}^b$ by sampling uniformly (with or without replacement) $K$ times from $\mathcal{D}$
    Compute model weights $\hat{w}^b = \mathcal{A}(\mathcal{D}^b)$
  **end for**
**output** Ensemble weights $\hat{w} = \frac{1}{B} \sum_{b=1}^{B} \hat{w}^b$

---

We decompose the model selection procedure $\mathcal{M}$ into two stages

$$\mathcal{M} = \mathcal{S} \circ \mathcal{A}, \tag{1}$$

where $\mathcal{A}$ maps data $\mathcal{D}$ to weights $\hat{w} = \mathcal{A}(\mathcal{D}) \in \mathbb{R}^{|M^+|}$, where for each $m \in M^+$, $\hat{w}_m$ is the weight assigned to model $m \in M^+$. This weight may be loosely interpreted as the probability of the candidate model being the best model given the data among the choices in $M^+$. The selection rule $\mathcal{S}$ takes these weights $\hat{w}$ as input and returns a non-empty set of selected model(s) $\hat{M} \subseteq M^+$ that provide the best performance as a function of the training data, where performance may reflect some combination of fit to data and structural properties of the model.

## 2.1 Stage 1: Assigning weights to candidate models with $\mathcal{A}$

**Base algorithm $\mathcal{A}$.** A base algorithm $\mathcal{A}$ is any off-the-shelf model weighting algorithm. Many model selection methods lead to weight vectors $\hat{w}$ that place all weight on a single model (i.e., the algorithm $\mathcal{A}$ returns only one model, so $w$ is a vector of zeros with an entry of 1 corresponding to the selected model). Examples of such algorithms are LASSO for variable selection (Tibshirani, 1996), SINDy for model discovery for dynamical systems (Brunton et al., 2016), and graphical LASSO for graph subset selection (Friedman et al., 2008), all of which are experimentally explored in this work. A known issue of these procedures is that these algorithms can produce unstable model selections. One mechanism that has been shown to help its stability is bagging.

**Bagged algorithm $\tilde{\mathcal{A}}_{K,B}$.** We denote $\tilde{\mathcal{A}}_{K,B}$ as the algorithm resulting from applying a bagging technique on a base algorithm $\mathcal{A}$, where $K$ is the number of samples per bag and $B$ is the number of bags. The output of the weighting algorithm $\tilde{\mathcal{A}}_{K,B}$ must be bounded. For simplicity, we assume that the weights can be normalized such that $\hat{w}$ belongs to the probability simplex $\Delta_{|M^+|-1}$. [2] Bagging averages an ensemble of weight vectors fit on different randomly sampled bags; see Algorithm 1. Sampling with replacement is known as *bagging*, and sampling without replacement is known as *subbagging*.

For a base algorithm $\mathcal{A}$ that places all weight on a single model, bagging $\mathcal{A}$ to compute $\hat{w}_m$ simply counts the fraction of bags where $m$ was selected. Even if the set of candidate models is infinite (i.e., $|M^+| = \infty$), the weights $\hat{w}$ will have at most $B$ nonzero entries, so bagging is still tractable. Generally, increasing the number of bags $B$ used in $\tilde{\mathcal{A}}_{K,B}$ provides a more stability, as we empirically show in Figures 12 and 13.

## 2.2 Stage 2: Conventional selection rules $\mathcal{S}$

We discuss conventional selection approaches $\mathcal{S}$ and their limitations.

**The standard argmax.** The default selection rule selects the best model(s) $\hat{M} \in \mathrm{argmax}_m \, \hat{w}_m$. ($\hat{M}$ may contain more than one model in the case of exact ties for the top weight.) If the weights contain near-maximizers, the argmax can be sensitive to small perturbations of $\hat{w}$, meaning that a different model may be selected with a small perturbation to the training data.

**Top-$k$.** Instead of returning the very best model(s) using the argmax, an alternative is to return the top $k$ models that receive the largest weight, where $k$ is user-specified. Similar to the argmax, top-$k$ fails to

---

[2]The probability simplex $\Delta_{|M^+|-1}$ consists of nonnegative weights $w$ that sum to one: $\sum_{m \in M^+} w_m = 1$.

automatically adapt to the inherent uncertainty in the model selection process. In particular, if $\hat{w}$ has far more than $k$ near-maximizers, top-$k(\hat{w})$ will be unstable. On the other hand, if $\hat{w}$ has fewer than $k$ near-maximizers, top-$k$ returns a set that is unnecessarily large.

**Stability selection.** In the context of variable selection, the stability selection method (Meinshausen and Bühlmann, 2010) selects a single model based on whether each covariate is included in the model with probability at least $\tau$. Concretely, consider a vector of weights $\hat{w}$ associated with candidate models $M^+$, where $|M^+| = 2^{[d]}$ since each of the $d$ covariates is either included or not for each model in $M^+$. Define the marginal *variable* probabilities $\hat{\Pi}_j = \sum_{m \subseteq [d]: j \in m} \hat{w}_m$, for $j = 1, \ldots, d$, where $j \in m$ corresponds to variable $j$ being included in model $m$. In other words, $\hat{\Pi}_j$ corresponds to the frequency of variable $j$ being included in any selected model. Stability selection returns $\mathrm{ip}_\tau(\hat{w}) := \left\{ j \in [d] : \hat{\Pi}_j \geq \tau \right\}$ for some use-specified threshold $\tau$. Similar to the standard argmax (without ties), $\mathrm{ip}_\tau(\hat{w})$ only returns a collection of *variables*, which together constitute a single *model*. This selection procedure can be unstable if some marginal probabilities $\hat{\Pi}_j$ are near the threshold $\tau$. §A presents a more detailed comparison between this method and our method, particularly in the setting of correlated variables.

### 2.3 Model selection stability

The stability of $\mathcal{M}$ corresponds to how much the output of $\mathcal{M}$ may change with small perturbations to $\mathcal{D}$ *for any dataset $\mathcal{D}$*. Note that this cannot be estimated empirically because we cannot test all possible datasets $\mathcal{D}$ with finite computational resources. We adapt the definition of stability in Soloff et al. (2024b) to the context of model selection.

**Definition 2.1.** A procedure $\mathcal{M}$ has *model selection stability* $1 - \delta$ at sample size $n$ if, for all datasets $\mathcal{D}$ with $n$ samples,

$$\frac{1}{n} \sum_{i=1}^{n} \mathbb{1}\{\hat{M} \cap \hat{M}^{\backslash i} = \varnothing\} \leq \delta, \tag{2}$$

where $\hat{M} = \mathcal{M}(\mathcal{D})$ and $\hat{M}^{\backslash i} = \mathcal{M}(\mathcal{D}^{\backslash i})$ for each $i \in [n]$.

In Definition 2.1, $\mathcal{D}^{\backslash i}$ refers to dataset $\mathcal{D}$ with the $i$-th sample removed. In plain language, the inequality in (2) quantifies stability in terms of whether the sets of selected models overlap when we drop a single observation at random. Note that this notion of model selection stability holds for any "black box" model selection method of the form (1), in contrast to prior works that focus on particular settings, such as variable selection (e.g., Meinshausen and Bühlmann (2010)) or $\kappa$-means clustering (e.g., Ben-David et al. (2007)). While this definition of stability may seem weak (i.e., in practice, much larger perturbations to the data could be made), we will see empirically in our results that this minimal change to the data can have a substantial impact on model selection stability. Moreover, stability according to this criterion is necessary for most stronger notions of stability to be satisfied.

## 3 Related work

Model selection has a rich history in the statistics literature. Ding et al. (2018) provides a recent overview of model selection techniques. Various aspects of model selection have been studied in previous work, including hyperparameter selection via cross-validation (Raschka, 2020), preventing overfitting (Cawley and Talbot, 2007), selection criteria (Rao et al., 2001), among others. In this work, we are particularly interested in the aspect of model selection stability, and bagging has been a key tool to address instability in model selection.

**Returning a model selection set.** Returning a set of models to acknowledge inherent uncertainty in model selection is an old idea. Early literature on subset selection for linear regression (Spjøtvoll, 1972) focused on selecting the true model with high probability. Breiman (2001) coined the terms "Rashomon set" to refer to a complete set of nearly equally performing models (e.g., Xin et al. (2022) finds the full Rashomon set for decision trees), and the "Rashomon effect" to refer to the phenomenon that multiple, vastly distinct models may all perform equally well on out-of-sample data for a particular problem. More recent works extend the idea of a *model confidence set* to more general modeling settings (Hansen et al., 2011; Zhang

et al., 2024). These works have a stochastic model for the data and construct a confidence set containing the best-fitting model with high probability. By contrast, our framework places no stochastic assumptions on the data, instead targeting a more modest goal of stability for the selected set.

**Stability in variable selection.** There is extensive literature on stability in variable selection; see, e.g., Khaire and Dhanalakshmi (2022) for a recent overview. Work in this area has overwhelmingly focused on "simple" procedures, returning a single subset of variables, and stability is measured based on the similarity of the selected subsets of variables. By contrast, Definition 2.1 is not satisfied by simple procedures unless they return the same model most of the time.

**Bagging to improve stability.** Bagging is an important tool for model selection and variable selection, and has been widely applied to stabilize model selection procedures (Breiman, 1996a;b). For example, *stability selection* thresholds the marginal bootstrap inclusion probabilities of each variable (Meinshausen and Bühlmann, 2010; Shah and Samworth, 2013). Additionally, *BayesBag* (Bühlmann, 2014) directly averages posterior distributions over different resamplings of the data. Huggins and Miller (2023) analyze the accuracy and stability properties of the BayesBag approach specifically for Bayesian model selection. Soloff et al. (2024a;c) shows that bagged weights are stable, meaning that dropping a single sample cannot change the bagged weights $\hat{w}$ by much in the Euclidean norm. Bagging can thus express model uncertainty, but, as we show empirically in §5, selecting the most frequent model among multiple close contenders can still be quite unstable. In order to achieve any stability guarantees of the selected model(s), a new selection rule $\mathcal{S}$ is needed, as discussed in §4.1.

**Stabilizing classification.** Our work builds on ideas from stable classification. Soloff et al. (2024b) introduces a framework for set-valued classification, combining the inflated argmax with bagging to guarantee stability for classification.

## 4 Our approach

We now provide a brief overview of our pipeline for stabilizing a model selection procedure $\mathcal{M} = \mathcal{S} \circ \mathcal{A}$.

1. First, we bag the weighting algorithm $\mathcal{A}$, i.e., average the weights fit on different random samples from $\mathcal{D}$. We fully define bagging in Algorithm 1.

2. Next, we pass the bagged weights $\hat{w}$ through the *inflated argmax*, which selects a set of near-maximizers of $\hat{w}$ that is robust to small perturbations of the weights. We define the inflated argmax for model selection in Definition 4.1.

In §4.2, we provide a stability guarantee for this pipeline that can be applied to any algorithm $\mathcal{A}$ and any data set $\mathcal{D}$. We first discuss our pipeline in detail.

### 4.1 The inflated argmax for model selection

In this section, we propose an alternative selection procedure $\mathcal{S}$ based on the inflated argmax. Soloff et al. (2024b) introduced the inflated argmax to guarantee stability in the context of multiclass classification. A key insight of this paper is that **if we formulate model selection as a multi-class classification problem, where each class corresponds to one of the candidate models, then we can directly use the inflated argmax and inherit its corresponding theoretical stability guarantees for model selection.** With this insight, the inflated argmax, an alternative model selection criteria $\mathcal{S}$ that we denote by $\mathrm{argmax}^\varepsilon$, will allow us to decide which subset $\hat{M}$ of $M^+$ to return.

**Definition 4.1** (Inflated argmax)**.** For $m \in M^+$, let

$$R_m^\varepsilon = \left\{ w \in \Delta_{|M^+|-1} : w_m \geq \max_{m' \neq m} w_{m'} + \frac{\varepsilon}{\sqrt{2}} \right\}, \tag{3}$$

where $\Delta_{|M^+|-1}$ is a probability simplex. For any $w \in \Delta_{|M^+|-1}$ and $\varepsilon > 0$, the inflated argmax is defined as

$$\mathrm{argmax}^\varepsilon(w) := \left\{ m \in M^+ : \mathrm{dist}(w, R_m^\varepsilon) < \varepsilon \right\}, \tag{4}$$

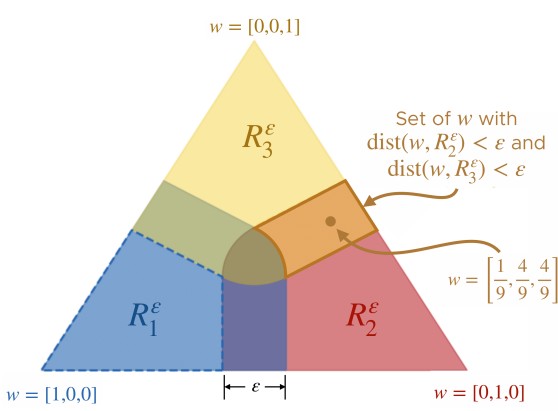

Figure 1: Visualization of argmax$^\varepsilon$ when $|M^+| = 3$. The weights assigned to each model $\mathcal{A}$ can be mapped onto a 3-dimensional probability simplex as shown. The example weight vector $w = \left[\frac{1}{9}, \frac{4}{9}, \frac{4}{9}\right]$ lands in an area that is within $\varepsilon$ distance of regions $R_2^\varepsilon$ and $R_3^\varepsilon$, which are the regions corresponding to returning models $m_2$ and $m_3$, respectively. Because this weight vector is in an "uncertain" region based on $\varepsilon$ for distinguishing whether model $m_2$ or $m_3$ provides a better fit to the data, argmax$^\varepsilon$ will return both models: $\{m_2, m_3\}$. In contrast, weight vector $w = [\frac{1}{9}, \frac{1}{9}, \frac{7}{9}]$ would land in the area labeled $R_3^\varepsilon$, and argmax$^\varepsilon$ will return $m_3$ only.

where $\text{dist}(w, R_m^\varepsilon) = \inf_{q \in R_m^\varepsilon} ||q - w||$, and $|| \cdot ||$ is the Euclidean norm.

A visualization of a simple example of argmax$^\epsilon$ where more than one model is returned is shown in Figure 1. An intuitive explanation of Definition 4.1 is that $m \in \text{argmax}^\varepsilon(w)$ indicates that $m$ would have the largest weight in $w$ by some margin if the weights $w$ were slightly perturbed, where both the size of the margin and the weight perturbations scale with $\varepsilon$. This definition naturally allows for the number of returned models to adapt to the uncertainty in the model selection: for example, if the top two competing models are close in probability, then the inflated argmax will return both models for some choice of small $\varepsilon$. On the other hand, if the top two competing models have a large enough gap in their two weights, the inflated argmax will only return the top model for a choice of small $\varepsilon$. The parameter $\varepsilon$ is *not* something to be tuned. It is chosen to satisfy a user-specified stability tolerance; see §4.2.

## 4.2 Stability guarantee

In this section, we state a theorem guaranteeing the stability of our approach based on our definition of stability in Definition 2.1.

**Theorem 4.2.** *(Adapted from Soloff et al. (2024b, Theorem 17).) For any model selection procedure $\mathcal{S} \circ \mathcal{A}$, our method* argmax$^\varepsilon \circ \widetilde{A}_{K,B}$ *satisfies model selection stability at instability level*

$$\delta = \frac{1}{\varepsilon^2}\left(1 - \frac{1}{|M^+|}\right)\left(\frac{1}{n-1}\frac{\rho}{1-\rho} + \frac{16e^2}{B}\right),$$

*where $\rho = \frac{K}{n}$ for subbagging and $\rho = 1 - (1 - \frac{1}{n})^K$ for bagging.*

The proof immediately follows from the proof presented in Soloff et al. (2024b). **The main use for this theorem is that the user can choose a tolerable worst-case instability $\delta$ for their model selection, input the known parameters of the problem** (i.e., $|M^+|$ candidate models, $K$ samples per bag, $n$ total samples, and $B$ bags)**, and directly solve for the $\varepsilon$ that should be used for argmax$^\varepsilon$.**

Recall that $\varepsilon$ determines how much we inflate the argmax, so when it's smaller, the inflated argmax is closer to the standard argmax. Accordingly, when $\varepsilon$ is smaller, Theorem 4.2 gives a larger $\delta$, reflecting a weaker stability guarantee. Further, note that the instability level $\delta$ decreases (i.e., stability increases) by decreasing the bag size $K$ and increasing the number $B$ of bags. The size of our model class $|M^+|$ has only a small influence on the level of instability of our procedure. For example, when there are two models, $|M^+| = 2$, the factor $1 - \frac{1}{|M^+|} = \frac{1}{2}$, giving a constant-factor boost in stability over the case where $M^+$ is countably infinite, where $1 - \frac{1}{|M^+|}$ becomes 1. This theoretical result is the first to provide theoretical guarantees for model selection stability that is agnostic to the underlying models, base algorithm, and data distribution.

| NOTATION | SELECTION CRITERION |
|---|---|
| $\text{argmax} \circ \mathcal{A}$ | Model(s) with maximum weight assigned by $\mathcal{A}$ |
| $\text{argmax} \circ \tilde{\mathcal{A}}_{K,B}$ | Model(s) with maximum weight assigned by $\tilde{\mathcal{A}}_{K,B}$ |
| $\text{top-}k \circ \tilde{\mathcal{A}}_{K,B}$ | $k$ models with $k$ largest weights assigned by $\tilde{\mathcal{A}}_{K,B}$ |
| $\text{ip}_\tau \circ \tilde{\mathcal{A}}_{K,B}$ | Model with variables with inclusion probability $\geq \tau$ in $\tilde{\mathcal{A}}_{K,B}$ |
| $\text{argmax}^\varepsilon \circ \tilde{\mathcal{A}}_{K,B}$ | $\varepsilon$-inflated argmax of $\tilde{\mathcal{A}}_{K,B}$ |

Table 1: Table of notation for each model selection algorithm. $\mathcal{A}$ refers to a base algorithm mapping a dataset $\mathcal{D}$ to empirical predicted probabilities $\hat{w}$ for each possible subset, and $\tilde{\mathcal{A}}_{K,B}$ is the subbagged version of $\mathcal{A}$ for bags with $K$ randomly selected (without replacement) samples.

## 5 Experiments

Our experiments assess and compare the stability of five different model selection approaches, described in Table 1. We perform experiments in three settings: identifying important variables in a synthetic dataset that contains a small collection of highly correlated variables (see Appendix C), identifying the latent Lotka-Volterra governing differential equations from noisy trajectory data generated from these equations, and constructing a graph to represent cell-signaling pathways of 11 proteins. In the first two settings, we generate synthetic datasets for $N$ trials, each contaminated with independent Gaussian noise, allowing us to examine the stability across a collection of datasets where we know the ground truth $m^*$.

In each experiment, we compute weights $\hat{w}$ across the set candidate models $M^+$ using a base algorithm $\mathcal{A}$ and a subbagged version of $\mathcal{A}$ denoted $\tilde{\mathcal{A}}_{K,B}$. Based on (2), we compute the stability for trial $j \in [N]$ using

$$\delta_j = \frac{1}{n} \sum_{i=1}^{n} \mathbb{1}\{\hat{M}_j \cap \hat{M}_j^{\setminus i} = \varnothing\}, \tag{5}$$

This empirical measure of stability measures, for trial $j$, the proportion of LOO selected models $\hat{M}_j^{\setminus i}$ for $i \in [n]$ that have no overlap with the set of selected models $\hat{M}_j$ returned by training with access to the entire dataset. We also compute the empirical cumulative distribution function (CDF)

$$\frac{1}{N} \sum_{j=1}^{N} \mathbb{1}\{\delta_j \leq \delta\}, \tag{6}$$

as a function of $\delta \in [0,1]$ to highlight variation across trials with different random seeds (shown in Figure 3 and Figure 8). Curves higher on the plot are better in the sense that the corresponding method achieves model selection stability $\delta$ (Definition 2.1) for a larger fraction of trials, and our primary interest is in small $\delta$ (i.e., high stability).

We additionally compute utility-weighted accuracy, inspired by Zaffalon et al. (2012), using

$$\frac{1}{N} \sum_{j=1}^{N} \frac{\mathbb{1}\{m^* \in \hat{M}_j\}}{|\hat{M}_j|}, \tag{7}$$

where $m^*$ is the "true" data generating model in the synthetic experiments. This measure down-weights the accuracy by the number of models returned $|\hat{M}_j|$. For example, if $|\hat{M}| = 1$, and the true model is contained in $\hat{M}$ (i.e., $m^* \in \hat{M}$), then both the accuracy and the utility-weighted accuracy are 100%. However, if $|\hat{M}| = 1,000$ and $m^* \in \hat{M}$, the accuracy is 100%, but the utility-weighted accuracy is 0.1%. The utility-weighted accuracy measure aims to identify returned model sets that are both accurate and interpretable, and we argue that a smaller number of returned models is more interpretable. This metric is used to compare methods in the center column of Figure 3.

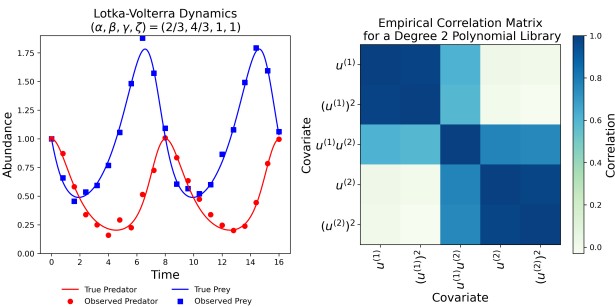

Figure 2: **(Left)** Lotka-Volterra trajectories for one simulated dataset. The time span ranges from 0 to 16, and there are 21 time points with predator-prey information contaminated with $\mathcal{N}(0, 0.05)$ distributed noise. **(Right)** Empirical correlation matrix for the variable library input into SINDy and E-SINDy. We use the library $(c, u^{(1)}, (u^{(2)})^2, u^{(1)}u^{(2)}, u^{(2)}, (u^{(2)})^2)$, where $c$ is a constant.

## 5.1 Model discovery of Lotka-Volterra dynamics

This experiment focuses on discovering Lotka-Volterra dynamics from data using SINDy. The Lotka-Volterra equations were first derived to model chemical reactions in the seminal work (Lotka, 1910) and later studied in the now classic use case of predator-prey dynamics (Lotka, 1925). The Lotka-Volterra equations are

$$\dot{u}^{(1)}(t) = \alpha u^{(1)}(t) - \beta u^{(1)}(t)u^{(2)}(t), \tag{8a}$$

$$\dot{u}^{(2)}(t) = -\gamma u^{(2)}(t) + \zeta u^{(1)}(t)u^{(2)}(t), \tag{8b}$$

where $u^{(1)}$ is the population density of prey, $u^{(2)}$ is the population density of predators, $\dot{u}^{(1)}$ and $\dot{u}^{(2)}$ are the population growth rates, and $t$ corresponds to time. The parameters $\alpha, \beta, \gamma, \zeta > 0$ respectively refer to the prey's growth rate, the predator's impact on the prey's death rate, the predator's death rate, and the prey's impact on the predator's growth rate (Lotka, 1925). We now imagine we do not know our system is governed by the Lotka-Volterra equations and wish to discover the governing model from data. Our concern is in recovering the correct terms rather than estimating the parameters.

### 5.1.1 Base algorithm $\mathcal{A}$: Sparse Identification of Nonlinear Dynamical Systems (SINDy) with Sequentially Thresholded Ridge Regression (STRidge)

One application of sparse regression is the discovery of governing equations of a system from observations. Specifically, we seek to learn the governing equations of a system of the form

$$\frac{d}{dt}u(t) = f(u(t)), \tag{9}$$

where $u(t) = \left[u^{(1)}(t), u^{(2)}(t), \cdots, u^{(d)}(t)\right]^{\top}$ is a $d$-dimensional state vector containing quantities of interest at time $t$, and $f$ is the function describing how the derivative of $u(t)$ evolves with time. The Sparse Identification of Nonlinear Dynamical Systems (SINDy, Brunton et al. (2016)) algorithm represents a differential equation of the form (9) as a weighted sum of terms in a library of functional forms,

$$\dot{u}(t) = \Theta(u)\Xi, \tag{10}$$

where $\Theta(u) \in \mathbb{R}^{n \times p}$ ($n$ samples of $p$ functionals) is the library of functionals of $u(t)$ (e.g., polynomial transformations), and $\Xi \in \mathbb{R}^{p \times d}$ are the fitted coefficients. $\Xi$ is assumed to be sparse, so our goal is to find which elements of $\Xi$ should be nonzero, which will immediately give the estimated structure of the governing equations. SINDy finds a sparse solution to $\Xi$ by solving a sparse linear regression problem; see §B for more details on sparse regression.

As a bagged extension of SINDy, Ensemble SINDy (E-SINDy, Fasel et al. (2022)) creates an ensemble of model fits, and with this ensemble, the final model is chosen via thresholding inclusion probabilities of each library term based on a tolerance $\tau$. This approach corresponds to applying stability selection (Meinshausen and Bühlmann, 2010; Shah and Samworth, 2013) to SINDy, using sequentially thresholded ridge regression (STRidge, Rudy et al. (2017)) instead of the LASSO for $\mathcal{A}$. E-SINDy aggregates the coefficients by selecting covariates whose inclusion probability is greater than some prespecified value $\tau \in (0, 1)$. Of the variables with an inclusion probability greater than $\tau$, the corresponding coefficients are set to the average fitted values across bags. The collection of variables with a nonzero coefficient correspond to the selected model. We refer to this ensemble approach as $\text{ip}_{\tau} \circ \tilde{\mathcal{A}}_{K,B}$, where the base algorithm $\mathcal{A}$ is SINDy solved with STRidge.

### 5.1.2    Data generation

We generate synthetic datasets containing $n = 21$ time points of abundance data for predator and prey populations, solve from the Lotka-Volterra equations and contaminate with Gaussian noise. This data is generated to match key characteristics of a real world predator-prey dataset from the Hudson Bay Company (Hewitt, 1921) in terms of the periodicity of observations in relation to the dynamics. $N = 100$ trials (i.e., independent datasets) are independently generated according to the same data generation process. We provide further details of this data generation process in §D.1. We visualize one set of trajectories in Figure 2.

To construct our library of functions, we selected degree-two polynomials as our covariates, and for one dataset, we visualize the corresponding covariate correlations in Figure 2. As seen in the correlation matrix, the covariate structure contains highly correlated variables (e.g., $u^{(1)}$ and $(u^{(1)})^2$), which impacts the model selection stability of SINDy. In our experiments, we solve the SINDy linear regression problem with STRidge penalization (Rudy et al., 2017) as our base algorithm $\mathcal{A}$, as in Fasel et al. (2022).

### 5.1.3    Results

We see that across various choices of $\varepsilon$, the inflated argmax provides enhanced stability while at the same time yielding small and accurate sets of selected model(s). This boost in utility-weighted accuracy is due to the adaptivity of the inflated argmax: this method will output a single model as often as possible, whereas for example, top-$k$ will always output $k$ model selections, leading to a comparatively lower utility in many cases. This point is further confirmed in the plot of the median number of models returned vs. worst-case instability in Figure 3. For a fixed level of worst-case instability empirically seen, we can visualize the empirical CDF to assess the distribution of instability values $\delta_j$ across trials $j \in [N]$. We see in Figure 3 that the inflated argmax provides a distribution of instability values much closer to 0 (i.e., higher stability) compared to all other conventional approaches investigated.

In summary, Figure 3 shows that (a) selection methods that return a single model (argmax and $\text{ip}_\tau$, even with subagging the base algorithm) are less stable than methods that return one or more selected models (i.e., the inflated argmax and top-$k$), (b) the inflated argmax provides higher utility-weighed accuracy scores for similar worst-case stability levels as top-$k$, meaning that the inflated argmax is returning smaller sets containig the correct model more often and (c) the inflated argmax provides smaller sets of returned models compared to top-$k$ for a wide range of worst-case stability values. We additionally plot the stability across the parameter $B$ in §E.

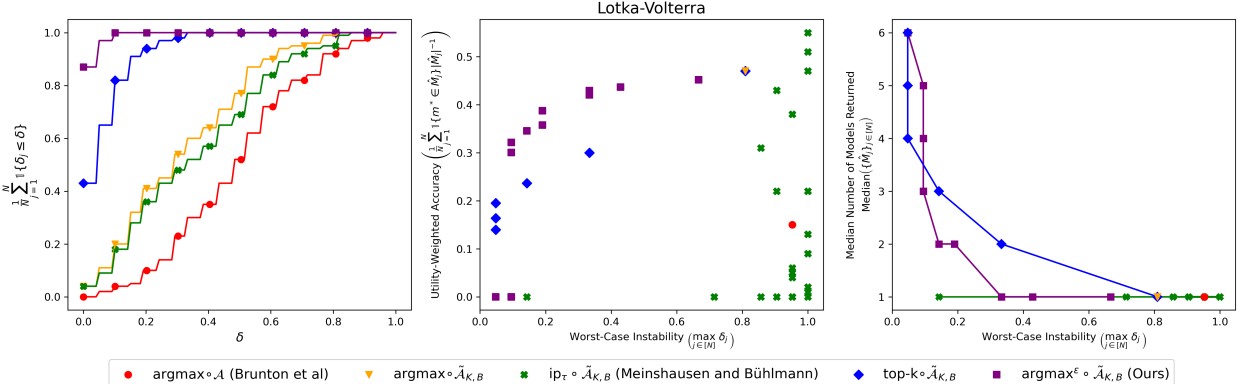

Figure 3: Lotka-Volterra results. **(Left column)** Empirical CDF for the stability measures $\delta_j$, computed in (5), across $j \in [N]$ for each $\mathcal{S}$. We chose the parameters $\tau = 0.63$, $k = 2$, and $\varepsilon = 0.09$ since these values yield a utility-weighted accuracy of approximately 0.3 for each $\mathcal{S}$. **(Center column)** Utility-weighted accuracy, computed in (7), versus the worst-case instability across $\delta_j$. **(Right column)** Median number of models returned across $j$ versus the maximum $\delta_j$ across $j \in [N]$ for each $\mathcal{S}$. **(Center and right columns)** We plot the range of values $k \in [1, \ldots, 6]$, and $\epsilon \in (0, 1)$ and $\tau \in (0, 1)$ with approximately evenly spread values across their supports.

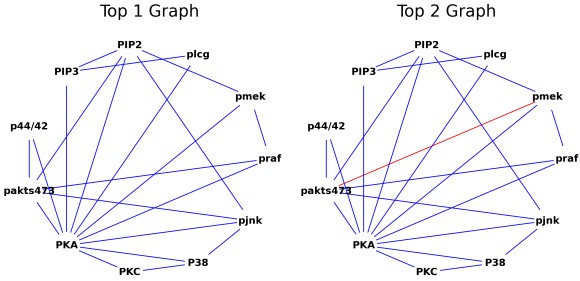

Figure 5: Visualization of the top two graph structures selected via subbagged gLASSO with 10,000 bags. The red connection shown in the right graph highlights that this connection is the only difference between the top 1 and top 2 selected graph structures. The top 1 graph was selected for 9.31% of bags, and the top 2 graph was selected for 8.54% of bags.

## 5.2 Graph selection on flow cytometry data

We consider a model selection problem where the model to select is the sparse connection structure within a graph. The base algorithm $\mathcal{A}$ we consider for this problem is the Graphical LASSO (gLASSO), which is developed in Friedman et al. (2008). The gLASSO estimates the structure of a graph by performing a

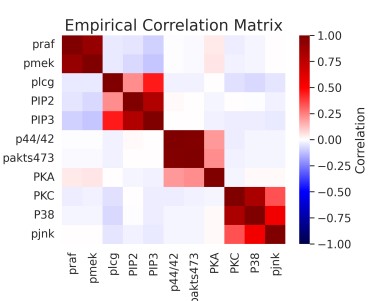

maximum likelihood estimation of the precision matrix $\Theta$ subject to an $l_1$ penalty of the estimated precision matrix. More precisely, the gLASSO assumes the data $x_i \sim \mathcal{N}(0, \Theta^{-1})$, where $x_i \in \mathbb{R}^d$ across samples $i \in [n]$, and finds an estimate $\hat{\Theta}$ such that

$$\hat{\Theta} = \underset{\Theta}{\operatorname{argmax}} \ \log \det \Theta - \operatorname{tr}(\tilde{\Sigma}\Theta) - \lambda ||\Theta||_1, \tag{11}$$

where $|| \cdot ||_1$ is the entry-wise $l_1$ norm, $\operatorname{tr}(\cdot)$ is the trace, $\tilde{\Sigma}$ is the sample covariance matrix, and $\lambda$ is the penalization hyperparameter. In our experiments, we select $\lambda$ via cross-validation, as described in §F, and hold this quantity fixed across our experiments.

Figure 4: Empirical correlation matrix of the 11 proteins. Data from Sachs et al. (2005).

We compute LOO stability results from a flow cytometry dataset in Sachs et al. (2005) (labeled "cd3cd28icam2+u0126"), containing $d = 11$ proteins (nodes) and $n = 759$ cells (samples). Figure 4 shows the empirical correlation structure of these 11 proteins in this dataset, which clearly shows grouped relationships among proteins.

| SELECTION METHOD | LOO INSTABILITY | AVG. LOO SET SIZE |
|---|---|---|
| $\operatorname{argmax} \circ \mathcal{A}$ | 0.453 | 1.00 |
| $\operatorname{ip}_{0.5} \circ \tilde{\mathcal{A}}_{K,B}$ | 0.013 | 1.00 |
| $\operatorname{argmax} \circ \tilde{\mathcal{A}}_{K,B}$ | 0.112 | 1.00 |
| $\operatorname{top-2} \circ \tilde{\mathcal{A}}_{K,B}$ | 0.008 | 2.00 |
| (Ours) $\operatorname{argmax}^{0.02} \circ \tilde{\mathcal{A}}_{K,B}$ | 0.008 | 1.58 |

Table 2: Table of stability and the average number of model structures returned across LOO trials for selecting the structure of the inverse precision matrix for a flow cytometry dataset taken from Sachs et al. (2005). In this experiment, $\mathcal{A}$ is the graphical LASSO (Friedman et al., 2008), $\tilde{\mathcal{A}}_{K,B}$ is the bootstrapped graphical LASSO with $B = 10,000$ and $K = 700$.

### 5.2.1 Results

Since we have one dataset to compute the LOO stability, $N = 1$. We compute stability results for the algorithms gLASSO ($\mathcal{A}$) and subbagged gLASSO ($\tilde{\mathcal{A}}_{K,B}$), where $K = 700$ and $B = 10,000$, for the selection methods argmax, $\operatorname{ip}_{0.5}$, top-2, and $\operatorname{argmax}^{0.02}$. The value for $\varepsilon$ for the inflated argmax was chosen so that the LOO instability closely matched the LOO instability of top-2, and $\tau$ in $\operatorname{ip}_\tau$ was chosen so that the number of selected connections was approximately the the same as the number of connections given by the top selected subbagged graph. These results are shown in Table 2.

As seen in Table 2, our method, argmax$^{0.02}$, provides the lowest LOO instability while minimizing the average number of graphs selected across the 759 LOO model fits. We also note that the percentage of model fits on bags that resulted in the top 2 models shown in 5 are relatively close: 9.31% for the top graph structure versus 8.54% for the second most selected graph structure, which suggests a reasonable degree of uncertainty in the "best" model. Due to this uncertainty in the selected model, selection methods that return only one model (i.e., argmax and ip$_\tau$) cannot sufficiently account for this uncertainty in the top model. Since the top two graph structures differ by one connection, this kind of result can provide evidence for follow up experiments, for example, to elucidate the relationship between the pakts473 and pmek proteins.

Though there is no agreed-upon "ground truth" graph connection structure for this real world example, we can compare the identified connections between proteins with those established in the literature. For example, Sachs et al. (2005) visualizes connections between proteins based on scientific evidence, which include PIP3-plcg, praf-pmek, PKA-praf, PIP2-PIP3, PKA-p38, and others, where some of these listed connections are mediated via intermediate proteins. These validated connections are seen in Figure 5. We note, however, that hyperparameter $\lambda$ in (11) largely impacts the accuracy by controlling the number of connections retained in the graph. In our experiment, this parameter was chosen in a purely data-driven way in §F, but a different value may have been chosen with domain expertise.

### 5.3 Selecting the number of clusters for $\kappa$-means

In this section, we present results on simulated data of an unsupervised $\kappa$-means[3] clustering (MacQueen, 1967) example where the task is to identify the number of clusters $\kappa$ in the data. This examples illustrates our algorithm's use on non-variable selection types of problems. Another example on decision trees can be found in §H. We generate $N = 100$ independent datasets, each with sample size $n = 30$ with a true number of clusters $\kappa = 3$. Further details of our synthetic data generation can be found in §G.

#### 5.3.1 Results

In this setting, each model corresponds to a different number of clusters $\kappa$, so the model selection problem is to choose $\kappa$. In our experiments, our base algorithm $\mathcal{A}$ is a deterministic computation of the "elbow" of the sum of squared distances versus number of clusters $\kappa$, which is a heuristic approach introduced in Thorndike (1953) and commonly used in unsupervised clustering tasks. For each $\kappa$, clusters are partitioned with the Lloyd-Forgy algorithm (Forgy, 1965), the standard algorithm for $\kappa$-means. Further details of our data generation and implemented methods can be found in §G.2. We construct $\tilde{\mathcal{A}}_{K,B}$ by bagging this base algorithm by computing $B = 10,000$ bags, each with $K = 25$ samples per bag. Figure 6 shows the main results of our experiment. We notably do not include results for the model selection method ip$_\tau \circ \tilde{\mathcal{A}}_{K,B}$ since that method is only applicable to variable selection problems.

The leftmost panel shows the empirical distribution of stabilities across methods, with fixed values of $\varepsilon = 0.3$ and $k = 2$ for argmax$^\varepsilon$ and top-$k$. Choosing $k = 2$ provides 100% stability, meaning there are seemingly only two reasonable choices for the number of clusters $\kappa$. The argmax$^{0.3}$ showed an empirical worst-case instability of 0.067 and an average returned set size of 1.18 across LOO trials, which provides high stability with minimal returned models. The middle plot of Figure 6 shows that our approach provides the best utility-weighted accuracy and stability compared to competing methods. In this setting, accuracy is measured based on how often the true $\kappa = 3$ is returned in $\hat{M}_j$ across all $j = 1, \ldots, N$ trials). This result shows that the inflated argmax returns sets that are both accurate and as small as possible at each fixed $\varepsilon$ value. This point is further confirmed by looking at the rightmost plot of the median number of returned models versus worst-case instability. Across a wide range of choices of worst-case instabilities, the inflated argmax consistently returns interpretable sets.

---

[3]In this paper, we use $K$ to represent the number of samples in a bag, $k$ to correspond to the number of models returned by a top-$k$ selection procedure, and $\kappa$ to represent the number of clusters in a clustering problem.

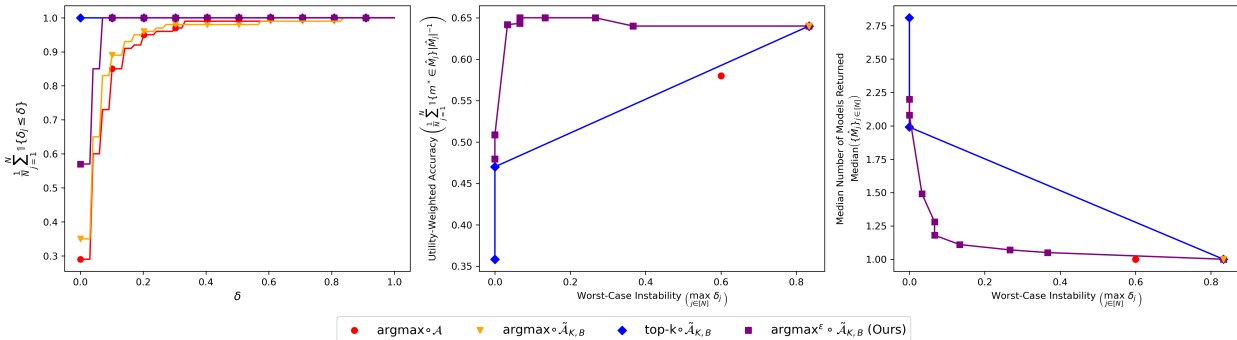

Figure 6: Unsupervised $\kappa$-means clustering results. **(Left column)** Empirical CDF for the stability measures $\delta_j$, computed in (5), across $j \in [N]$ for each $\mathcal{S}$. We chose the parameters $k = 2$, and $\varepsilon = 0.3$. **(Center column)** Utility-weighted accuracy, computed in (7), versus the worst-case instability across $\delta_j$. **(Right column)** Median number of models returned across $j$ versus the maximum $\delta_j$ across $j \in [N]$ for each $\mathcal{S}$. **(Center and right columns)** We plot the range of values $k \in [1, 2, 3]$, and $\epsilon \in [0.005, 0.01, 0.05, 0.1, 0.2, 0.3, 0.5, 0.7, 0.95, 1.0]$.

## 6 Discussion

Our framework provides multiple opportunities for further investigation, including exploring the interplay of stable model selection and identifiability, as well as stabilizing model selection without bagging, therefore reducing computation. Moreover, this work primarily focuses on providing theoretical stability guarantees for model selection, but in practice, a practitioner would also be interested in maximizing model performance. Additional future work could explore maximizing performance subject to stability guarantees.

We believe another interesting direction is developing similar theoretical stability results for other measures of stability aside from Definition 2.1. Our definition of stability is relatively weak, reflecting a bare minimum criterion necessary (though not sufficient) for many other notions of stability. Moreover, a limitation of our stability measure is that it may favor larger sets since two returned sets need to overlap by *only* one model to achieve a score of 1 for that LOO trial. For example, if $|\hat{M}_j|$ and $|\hat{M}_j^{\backslash i}|$ are large but $|\hat{M}_j \cap M_j^{\backslash i}| = 1$, the LOO stability for trial $i$ would be 1, which may not reflect what one would colloquially deem as "stable." An interesting alternative stability measure to theoretically analyze would be one that computes the percent overlap between elements of $\hat{M}_j$ and $\hat{M}_j^{\backslash i}$.

A natural follow up question to our approach is what a practitioner should do if multiple models are returned. As noted in §5.2.1, top models that substantially overlap can prompt directed follow-up experiments. Moreover, the size of the set of returned models may be a useful indicator of model uncertainty for a practitioner. The decision for how to handle multiple models should be carefully considered and align with the practitioner's ultimate goals (e.g., model interpretation or predictive accuracy).

The combination of bagging and the inflated argmax described in this paper can be applied to any black-box model selection procedure and offer stability guarantees described in Theorem 4.2. We define a notion of stability that is distinct from other works, one that computes the proportion of LOO selected models that are disjoint from the selection made by training with access to the full data. This is in contrast to, for example, *covariate-level* stability in linear models.

Our experiments illustrate that, in addition to providing theoretical stability guarantees, our method outperforms *ad hoc* procedures. Specifically, choosing the most frequently selected model across bags (i.e., $\text{argmax} \circ \tilde{\mathcal{A}}_{K,B}$) provides limited stability gains relative to an unstable base algorithm. Selecting the top-$k$ models across bags enhances stability but is not adaptive to the underlying empirical uncertainty of the model selection process; $k$ models will always be returned to the user. Choosing the returned model based on marginal inclusion probabilities ($\text{ip}_\tau$) of individual covariates is limited to covariate stability tasks and presents hyperparameter selection challenges (i.e., how to select $\tau$). In contrast, the inflated argmax *adapts*

to the number of models returned based on the underlying uncertainty, provides an approach for empirically protecting against a worst-case instability, and provides competitive utility-weighted accuracies (i.e., returns the correct model in the smallest possible $\hat{M}$).

## Acknowledgments

We gratefully acknowledge the support of the National Science Foundation via grant DMS-2023109 and the NSF-Simons National Institute for Theory and Mathematics in Biology via grants NSF DMS-2235451 and Simons Foundation MP-TMPS-00005320. MA gratefully acknowledges the support of the NSF Graduate Research Fellowship Program under Grant No. DGE-1746045. JS gratefully acknowledges the support of the Office of Naval Research via grant N00014-20-1-2337, and the Margot and Tom Pritzker Foundation.

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

## A   Example: Linear regression with correlated covariates

Consider the setting in which we have a set of covariates $x = \left[ x^{(1)}, x^{(2)}, \cdots, x^{(d)} \right]^{\top}$ where

$$\mathbb{E}[Y|X = x] = x^{(1)} + x^{(3)}, \tag{12}$$

and we seek to learn this relationship using sparse variable selection. If $x^{(1)}$ is highly correlated with $x^{(2)}$, and $x^{(3)}$ is highly correlated with $x^{(4)}$ and $x^{(5)}$ (see covariance matrix in Figure 7), then subtle changes in the dataset could result in selecting any of the following models $m_k(\mathbf{x})$:

$$\begin{aligned}
m_1(\mathbf{x}) &= \{x^{(1)}, x^{(3)}\}, \quad m_2(\mathbf{x}) = \{x^{(1)}, x^{(4)}\}, \\
m_3(\mathbf{x}) &= \{x^{(1)}, x^{(5)}\}, \quad m_4(\mathbf{x}) = \{x^{(2)}, x^{(3)}\}, \\
m_5(\mathbf{x}) &= \{x^{(2)}, x^{(4)}\}, \quad m_6(\mathbf{x}) = \{x^{(2)}, x^{(5)}\}.
\end{aligned} \tag{13}$$

If each of the six models above were selected with approximately equal frequency across bags, the output of Algorithm 1, would be approximately

$$\hat{w} \approx (1/6) \left[ 1, 1, 1, 1, 1, 1, 0, 0, \cdots \right]. \tag{14}$$

A selection method that finds the argmax of $\hat{w}$ would be unstable despite the stability of $\hat{w}$ derived from bagging. We seek a method that outputs models corresponding to selected variable sets

$$\left\{ \{x^{(1)}, x^{(2)}\} \times \{x^{(3)}, x^{(4)}, x^{(5)}\} \right\}, \tag{15}$$

accurately indicating that our method suggests either $x_1$ or $x_2$ should be included in the selected model, but not necessarily both. In other words, this output reflects the fact that our model selection method cannot reliably distinguish among the six models in (13).

**Limitations of the argmax.** The argmax applied to $\hat{w}$ in (14) will be highly sensitive to small perturbations in $\hat{w}$. Furthermore, the single best model does not reflect the inherent uncertainty in the model selection and that several models are almost equally likely.

**Limitations of top-$k$.** If the user chooses $k = 2$, only 2 models will be returned to the user, which does not reflect that any of the 6 models are approximately equally likely. This failure to adapt to uncertainty also applies to the other direction: if the user chooses $k > 6$, top-$k$ will still give back $k$ models, despite the fact that only 6 are highly likely.

**Limitations of stability selection** The stability selection method (Meinshausen and Bühlmann, 2010) will select the model corresponding to selected variables $\{x^{(1)}, x^{(2)}, x^{(3)}, x^{(4)}, x^{(5)}\}$ if $\tau$ is sufficiently small. This result can be misleading, in that it ultimately selects a model with five selected covariates as opposed to indicating that we lack certainty about which of multiple two-covariate models is correct.

**Benefits of the inflated argmax.** For a large enough choice of $\varepsilon$, the $\text{argmax}^\varepsilon$ would return all 6 models in (13). This output can be reported as a Boolean expression as in (15), which accurately reflects the uncertainty in the model selection.

## B   Variable selection in linear models with LASSO

In the linear case, the data is modeled as the form

$$y_i = \langle x_i, \beta \rangle + \eta_i, \ \eta_i \sim N(0, \sigma^2), \tag{16}$$

where $\sigma^2 \in \mathbb{R}$ is a vector containing the response error variance, and $\beta \in \mathbb{R}^d$ is the vector of regression coefficients. In many practical settings, the true population $\beta$ is hypothesized to be sparse, and the goal is to accurately estimate which elements of $\beta$ are nonzero.

Consider the following common model selection approach: first solve the *least absolute shrinkage and selection operator* (LASSO, Tibshirani (1996)) problem

$$\hat{\beta} = \text{argmin}_\beta \left\{ \frac{1}{n} \sum_{i=1}^n (y_i - \langle x_i, \beta \rangle)^2 + \lambda R(\beta) \right\}, \tag{17}$$

where $\lambda$ is a hyperparameter controlling the sparsity of $\beta$, and the regularization term $R(\beta) = ||\beta||_1$. Then select variables via

$$\hat{m}(\mathcal{D}) = \mathbb{1}\{|\hat{\beta}| > 0\} \tag{18}$$

where $\hat{\beta}$ implicitly depends on the dataset $\mathcal{D}$. In this context, the LASSO provides unstable model selections, particularly in settings with nontrivial correlations among covariates (Meinshausen and Bühlmann, 2010). Moreover, since many variable selection algorithms, including the LASSO, only return a single set of selected covariates, these methods do not directly provide an approach for assessing uncertainty in the selected model.

## C   Experiment: Penalized linear regression experiment with highly correlated covariates

This experiment is based on the motivating example in (12), where we have two important covariates and a small collection of highly correlated covariates.

### C.1   Data generation

In our standard regression example, we simulate datasets for $j \in [N]$ trials, where $N = 200$, according to the following process:

$$x_i \sim \mathcal{N}(0, C), \tag{19}$$

where $x_i \in \mathbb{R}^d$, $\mathcal{N}$ denotes a Normal distribution, and $C \in \mathbb{R}^{d \times d}$ is visualized in Figure 7. Figure 7 visualizes the covariance structure of the 20 covariates in the design matrix $X$, where the nonzero covariance between distinct covariates is 0.99, corresponding to extremely correlated covariates. For a single trial $j$, we simulate a dataset with $d = 20$ covariates and $i \in [n]$ samples. We take the first and third columns as the covariates used to create the response $y$,

$$y_i = x_i^{(1)} + x_i^{(3)} + v_i, \quad v_i \sim \mathcal{N}(0, 0.3^2), \tag{20}$$

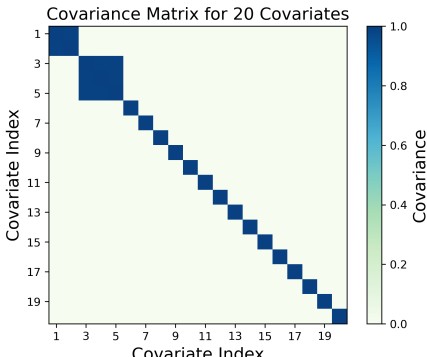

Figure 7: Covariance matrix for each simulated dataset. Covariate indices 1 and 3 were used to construct the response $y$.

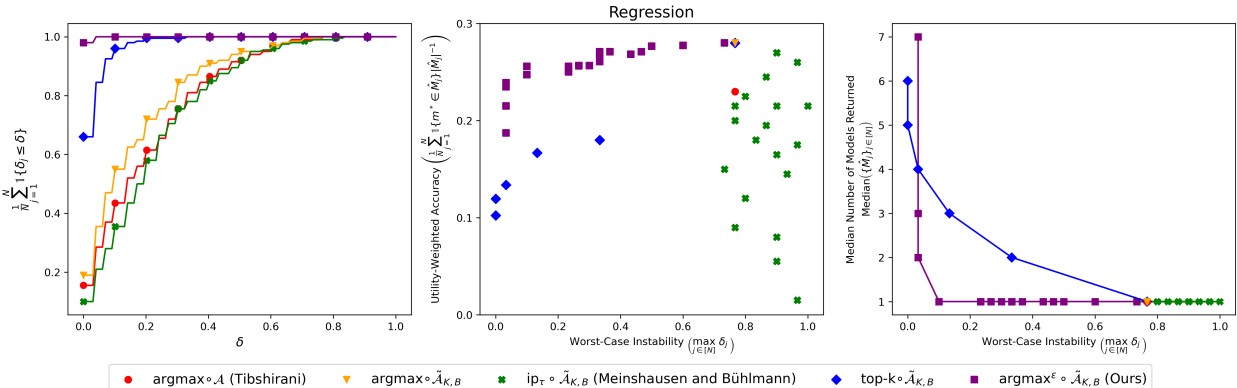

Figure 8: Regression results. In the left plot, we plot the stability curves for the parameters $\tau = 0.3$, $k = 2$, and $\varepsilon = 0.8$ since these values yield a utility-weighted accuracy of approximately 0.18 for each $\mathcal{S}$. **(Left column)** Empirical CDF for the stability measures $\delta_j$, computed in (5), across $j \in [N]$ for each $\mathcal{S}$. **(Center column)** Utility-weighted accuracy, computed in (7), versus the worst-case instability across $\delta_j$. **(Right column)** Median number of models returned across $j$ versus the maximum $\delta_j$ across $j \in [N]$ for each $\mathcal{S}$. **(Center and right columns)** We plot the values $k \in [1, \ldots, 6]$, and $\epsilon \in (0, 1)$ and $\tau \in (0, 1)$ with approximately evenly spread values across their supports.

where $\beta = [1, 1]^{\top}$, $x_i^{(1)}$ and $x_i^{(3)}$ are respectively the first and third covariates for sample $i$, and $\upsilon_i$ represents independent Gaussian distributed observation noise with mean 0 and standard deviation 0.3 for sample $i$.

We solve each optimization problem (i.e., fits on the full datasets $\mathcal{D}$, LOO datasets $\mathcal{D}^{\backslash i}$ for $i \in [n]$, and bags $\mathcal{D}^b$ for $b \in [B]$ across trials $j \in [N]$) for $\hat{\beta}$ using the optimization function in (17) for $R(\beta) = ||\beta||_1$, which is equivalent to the LASSO.

### C.1.1 Optimization hyperparameter

Selecting $\lambda$ via cross-validation leads to much denser models than the ground truth model, so we increase $\lambda$ to a level that induced more sparsity. This approach implicitly assumes we have prior knowledge of approximately how many coefficients are important in the regression task, but not necessarily which are important. We select $\lambda = 0.5$ since this penalization level led to highly sparse variable selections. We note, however, that the theory behind the inflated argmax is *model agnostic*, meaning that stability guarantees apply for any choice of $\lambda$.

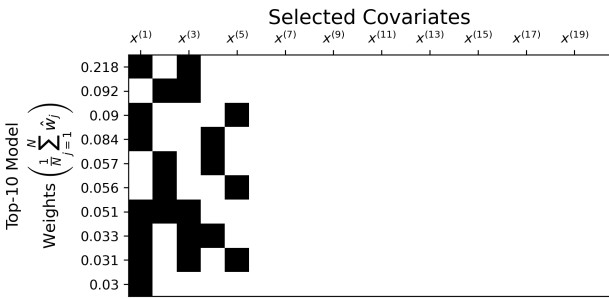

Figure 9: Top-10 selected models with their corresponding weights $\frac{1}{N}\sum_{j=1}^{N}\hat{w}_m^{(j)}$, which averages the weights across trials. Selected covariates for each model are shown with a black square.

## C.2 Small sample size ($n = 30$n=30)

We generate independent datasets (i.e., trials) for a small sample size, $n = 30$. For each of the $N = 200$ trials, we generate a dataset with $d = 20$ covariates and $n = 30$ training samples.

### C.2.1 Base algorithm $\mathcal{A}$A: LASSO

Our learning algorithm $\mathcal{A}$ is the LASSO, which corresponds to solving (17) with $R(\beta) = ||\beta||_1$, and $\tilde{\mathcal{A}}_{K,B}$ corresponds to a subbagged LASSO with $B = 10,000$ bags and $K = 25$ training samples per bag.

### C.2.2 Results

Figure 8 provides an empirical CDF of $\delta$ (6), the utility-weighted accuracy (7) versus the empirical worst-case instability, and the median number of sets returned across $N$ trials versus the worse-case instability. The empirical CDF across our chosen selection methods $\mathcal{S}$ show that the selection methods that return a single model, the argmax and $\text{ip}_\tau$, are the least stable. Subagging, which helps in stabilizing the estimated weights $\hat{w}$ assigned to each model $m$, yields marginal gains in stability when using the argmax. Top-$k$ and the inflated argmax, on the other hand, provide enhanced stability due to the fact that these methods are able to return more than one model. The $\text{ip}_\tau$ selection method fails to control the worst-case instability across a variety values of $\tau$, but is able to achieve good utility-weighted accuracy. Top-$k$ is able to achieve good utility-weighted accuracy with stable estimates. However, the inflated argmax is able to attain similar stability as top-$k$, while achieving a higher utility-weighted accuracy across a range of stability levels.

We confirm that the inflated argmax frequently outputs a single model by plotting the median number of models returned compared to the worst-case instability across various choices of $\varepsilon$ compared to various choices of $k$.

To illustrate that returning multiple models, when necessary, can be represented compactly, we borrow notation from logic to express an example of 6 models, reported in Figure 9. Figure 9 shows that the model weights averaged across trials $\left(\frac{1}{N}\sum_{j=1}^{N}\hat{w}^{(j)}\right)$. The top 6 models can be reported as

$$\left\{\{x^{(1)}, x^{(2)}\} \times \{x^{(3)}, x^{(4)}, x^{(5)}\}\right\}$$

which succinctly describes the model options.

## C.3 Larger sample size ($n = 300$n=300)

To illustrate the utility of the inflated argmax in a setting where this method is particularly useful, we construct an example where the model weights $\hat{w}$ estimated via bootstrapping suggest a high level of uncertainty in the model selection. We focus on one trial, so $N = 1$. In constructing this example, we follow a similar procedure as described in (19) and (20). We set $d = 200$, $n = 300$, $B = 10,000$, $K = 25$, $\lambda = 0.5$, and

$v_i \sim N(0, 0.5^2)$, where 0.5 is the response error standard deviation for each sample $i$. The correlation structure of covariates is extended to 200 covariates, where similarly the first 5 covariates are highly correlated with a correlation of 0.99 as in Figure 7, and $x^{(1)}$ and $x^{(3)}$ are used to construct $y$ with $\beta = [1, 1]^\top$.

Figure 10 shows the selections of among the first 5 covariates out of the 200 total for the top-10 models (covariates 6 through 200 are not selected in any of the top-10 models) using the full dataset versus a LOO dataset. As an illustrative example, we choose a LOO dataset that yielded a different top model compared to the top model using the full dataset.

The model weights clearly show uncertainty in the selection since the weights estimated for each of the top-10 models are very close together, suggesting that small perturbations of the dataset could impact the top model selected. The empirical stability for $\text{argmax} \circ \tilde{\mathcal{A}}_{K,B}$ using (5) yields $\delta_j = 0.14$ for this one trial $j$. This result means that 14% of the time in our 300 LOO trials, when a single sample was removed from the full dataset, a model other than $\{x^{(2)}, x^{(3)}\}$ (the model selected using all 300 samples) was selected.

The inflated argmax, on the other hand, will select more than one model in this instance for relatively small $\varepsilon$ due to the large uncertainty in the model weights. To concretely show this result, we compute the inflated argmax's model selection(s) and choose $\varepsilon$ using the result in Theorem 4.2. Setting $\delta$ to a fixed value, we can compute $\varepsilon$ via the following expression for subbagging

$$\varepsilon = \sqrt{\frac{1}{\delta} \left( \frac{1}{n-1} \frac{K/n}{1 - K/n} \right)}. \tag{21}$$

Since $|M^+| = 2^{200}$, we set $\frac{1}{|M^+|}$ to 0. Additionally, since the Monte-Carlo error term $\frac{16e^2}{B}$ is known to be overly conservative, and since we use a large number of bags $B$ in this experiment, we set this term equal to 0 as well. With $n = 300$, $K = 25$, and $\delta = 0.05$, we compute that $\varepsilon = 0.078$.

The instability of $\text{argmax}^{0.078} \circ \tilde{\mathcal{A}}_{K,B}$ is 0, meaning that for 0% of the 300 leave-one-out datasets, the inflated argmax returned a completely disjoint model selection. Additionally, the inflated argmax returned 6 models, which correspond to

$$\left\{ \{x^{(1)}, x^{(2)}\} \times \{x^{(3)}, x^{(4)}, x^{(5)}\} \right\}.$$

This set of returned models is very reasonable given the underlying correlation structure among the covariates.

Moreover, the type of stability in this work only measures changes the stability due to LOO changes to the dataset, which is a relatively small perturbation to the data. In reality, a method would ideally provide robustness to more substantial changes in the dataset (e.g., providing the same, or approximately the same, model selection(s) with a different dataset drawn from the same underlying distribution). The inflated argmax is better equipped than the argmax in this scenario by returning six models, reflecting the large uncertainty in selecting a model.

# D   Main experimental details

In our experiments, we utilize a cluster computing system to distribute parallel jobs across CPU nodes.

## D.1   Lotka-Volterra simulations

This section provides further details of the experiment in §5.1. We utilize the `pysindy` Python package (Kaptanoglu et al., 2022; de Silva et al., 2020) for their implementations of these methods to perform out Lotka-Volterra experiments with SINDy.

### D.1.1   Data generation

We generate data from the Lotka-Volterra differential equations described in (8) for $(\alpha, \beta, \gamma, \zeta) = (2/3, 4/3, 1, 1)$ for 21 evenly spread time points $t$ in the range $t \in [0, 16]$ and initial condition $[1, 1]^\top$. Solutions of this ordinary differential equation were obtained with the LSODA (Petzold, 1983) solver. For each

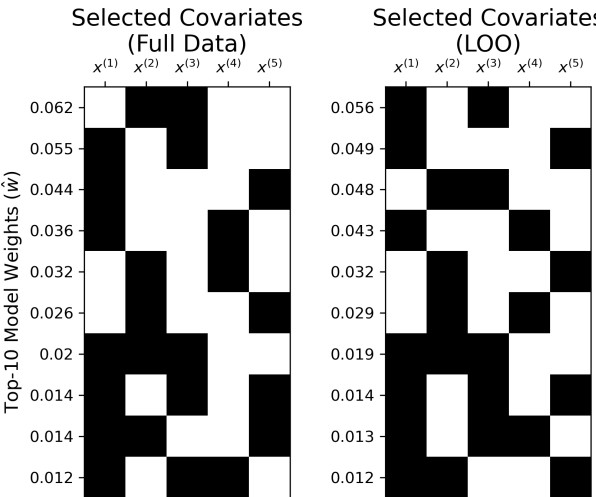

Figure 10: Models with the top 10 largest weights assigned via subbagging with $B = 10,000$ bags for one trial. The y-axis labels correspond to the model weights estimated with 10,000 bags. Covariates within each model (row) that were selected to have a nonzero estimated coefficient by LASSO are shown in black, and covariates that were not selected are shown in white. Only the first 5 covariates are shown since the top 10 models did not select covariates any of the covariates $x^{(6)}, x^{(7)}, \ldots, x^{(200)}$. **(Left)** Top-10 models using the full dataset where $n = 300$. **(Right)** Top-10 models using a LOO dataset where $n - 1 = 299$.

$t \in (0, 16]$, non-inclusive of the initial condition, we contaminate each $u^{(1)}(t)$ and $u^{(2)}(t)$ with $\mathcal{N}(0, 0.05^2)$ Gaussian noise, where 0.05 is the standard deviation of the measurement noise level.

To obtain sparse solutions of the regression problem and therefore estimate the governing equations, we solve the optimization problem with STRidge (Rudy et al., 2017), which iteratively (a) solves the equation in (17) with $R(\beta) = ||\beta||^2$, corresponding to solving Ridge regression (Hoerl and Kennard, 1970), and (b) thresholds $\hat{\beta}$ based on $\omega$, where $\omega$ is the minimum magnitude needed for a coefficient in the weight vector. Any coefficient with an estimated magnitude below $\omega$ is set to 0. This process is repeated until a convergence criterion is met.

### D.1.2 Cross-validating optimization hyperparameters

We perform a grid search across two parameters to find a combination that leads to a low validation MSE. The 5-fold validation MSE is measured as

$$\text{val-MSE}(\lambda, \omega; 5) = \frac{1}{5|\mathcal{V}(v)|} \sum_{v=1}^{5} \sum_{t \in \mathcal{V}(v)} ||\hat{u}(t; \lambda, \omega) - u(t)||^2, \tag{22}$$

where $\mathcal{V}(v)$ is the collection of time points included in the validation evaluation for fold $v \in 1, 2, \ldots, 5$; $\hat{u}(t; \lambda, \omega)$ is a vector of solutions, containing $\hat{u}^{(1)}(t; \lambda, \omega)$ and $\hat{u}^{(2)}(t; \lambda, \omega)$, to the model selected by SINDy using optimization parameters $\lambda$ and $\omega$ at time $t$; $u(t)$ is a vector of the observed values for $u^{(1)}(t)$ and $u^{(2)}(t)$ at time $t$. The trajectories were temporally split into 5 disjoint sets, where each set contains temporally adjacent time points.

We provide a table of validation MSEs averaged across the 5 folds in Table 3. Given our discretization of $\lambda$ and $\omega$ values, we choose $\omega = 0.18$ and $\lambda = 0.01$ and keep these values fixed in our experiments presented in Appendix C. Figure 11 visualizes the corresponding model selected by 5-fold cross validation. The best estimated model is

$$\dot{u}^{(1)} = 0.636u^{(1)} + -1.255u^{(1)}u^{(2)}, \tag{23}$$

$$\dot{u}^{(2)} = -0.839u^{(2)} + 0.833u^{(1)}u^{(2)}, \tag{24}$$

| $\lambda$ | $\omega$ | | | | | |
|---|---|---|---|---|---|---|
| | 0.16 | 0.17 | 0.18 | 0.19 | 0.20 | 0.25 |
| 0.0075 | 0.0538 | 0.0538 | 0.0539 | 0.0982 | 0.0963 | 0.1156 |
| 0.01 | 0.0538 | 0.0538 | **0.0538** | 0.0981 | 0.0963 | 0.1942 |
| 0.02 | 0.1285 | 0.1285 | 0.1287 | 0.1275 | 0.1256 | 0.4403 |
| 0.03 | 0.1285 | 0.1285 | 0.3160 | 0.3166 | 0.3202 | 0.3451 |
| 0.15 | 0.0788 | - | - | - | - | |
| 1 | - | - | - | 10.65 | - | 14.94 |

Table 3: Table of hyperparameter combinations and corresponding validation MSEs for choosing the STRidge $\lambda$ and $\omega$ hyperparameters. Any combination of hyperparameter values that led to a null model (selecting none of the covariates to include in the model) for any of the validation folds is reported as "-". We choose the hyperparameter combination that should give sparser models (larger $\lambda$ and larger $\omega$) in the case of tied validation MSE, which leads us to choose $\lambda = 0.01$ and $\omega = 0.18$ (bolded and highlighted).

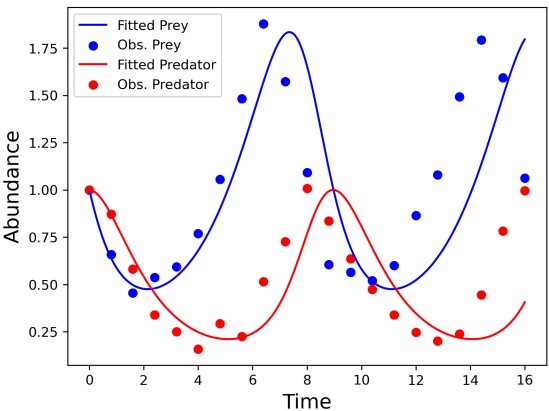

Figure 11: Lotka-Volterra solutions (solid lines) for the equations selected using SINDy with hyperparameters $\lambda = 0.01$ and $\omega = 0.18$ using one observational dataset (points).

which selects the correct terms in each differential equation with low error on the fitted $\hat{\beta}$ coefficients.

## E   Stability with varying numbers of bags: Lotka-Volterra example

The computational complexity of estimating the weights associated with each possible model $m \in M^+$ is not determined by the cardinality of the set $M^+$, but rather the number of bags $B$ used to estimated the weights associated with each model. Intuitively, we can generally expect that a larger number of bags (i.e., larger $B$) will correspond to a better estimate of the weights $\hat{w}$ for each model $m$ across selection methods. Figures 12 and 13 visualize the empirical CDF of instabilities $\delta_j$ across trials $j \in [N]$, where $N = 200$ in the regression experiments and $N = 100$ in the Lotka-Volterra experiment. A separate line is plotted for each choice of $B$.

In both plots, we see that $B = 10$ provides the least stability, evidenced by the larger proportion of $\delta_j$s near $\delta = 1$. The regression example provides evidence that generally, increasing the number of bags improves stability. However, this plot does not appear to clearly evidence this hypothesis as well as the Lotka-Volterra experiment. We hypothesize that since the covariates are highly correlated in the regression example with many reasonable models able to accurately predict the response, this setting may yield more variable stability results across the number of bags compared to the Lotka-Volterra example, which has a clear, unique model.

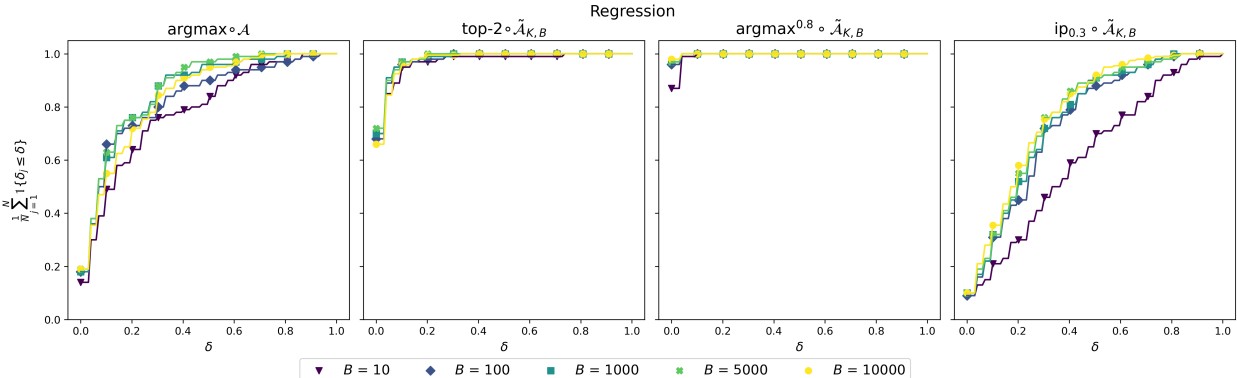

Figure 12: Empirical CDF of instability measures $\delta_j$ across trials $j \in [N]$ in the regression experiment for varying number of bags $B$ across four different selection methods: the $\mathrm{argmax}$, $\mathrm{top\text{-}2}$, $\mathrm{argmax}^{0.8}$, $\mathrm{ip}_{0.3}$. The values of $k$, $\varepsilon$, and $\tau$ were chosen to be consistent with the CDF plot in Figure 8.

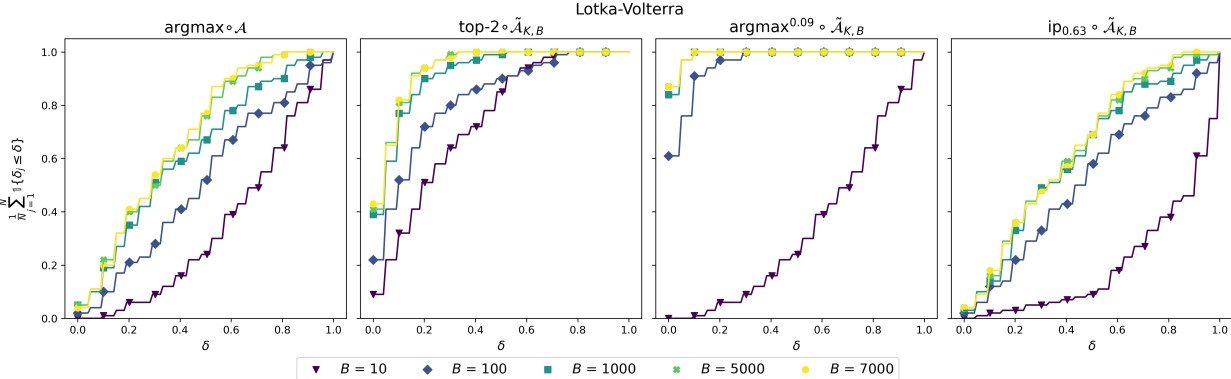

Figure 13: Empirical CDF of instability measures $\delta_j$ across trials $j \in [N]$ in the Lotka-Volterra experiment for varying number of bags $B$ across four different selection methods: the $\mathrm{argmax}$, $\mathrm{top\text{-}2}$, $\mathrm{argmax}^{0.09}$, $\mathrm{ip}_{0.63}$. The values of $k$, $\varepsilon$, and $\tau$ were chosen to be consistent with the CDF plot in Figure 3.

## F   Graph subset selection

### F.1   Cross-validation to select $\lambda$

We select the penalization hyperparameter $\lambda$ in (11) via 5-fold cross-validation. The validation log-likelihood averaged across the 5 folds is computed as

$$\text{val-log-likelihood}(\lambda; 5) = \frac{1}{5} \sum_{v=1}^{5} \left( \log \det \hat{\Theta}_v - \mathrm{tr}(\tilde{\Sigma}_v \hat{\Theta}_v) - \lambda ||\hat{\Theta}_v||_1 \right), \tag{25}$$

where $\hat{\Theta}_v$ is computed by solving (11) using the *training* data in fold index $v$, and $\tilde{\Sigma}_v$ is the sample covariance matrix computed from the *validation* data in fold index $v$. Computing (25) on the flow cytometry data across various choices of $\lambda \in [1, \dots, 500]$ yields the curve shown in Figure 14. Based on this figure, the best choice of $\lambda$ is $\lambda = 77$, which we keep constant throughout our experiments in §5.2.1.

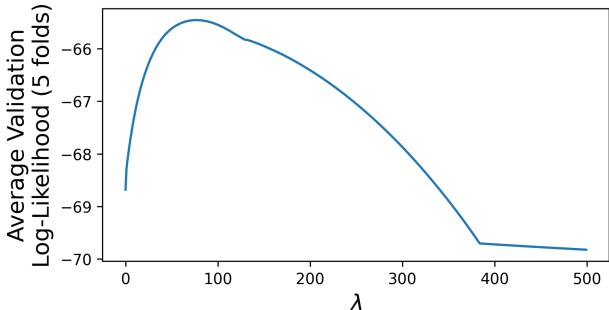

Figure 14: Average 5-fold validation log-likelihood across different choices of $\lambda$. The $\lambda$ that maximizes the log-likelihood averaged over 5 held-out validation folds is $\lambda = 77$.

# G $\kappa$-means clustering

## G.1 Data generation

We randomly generate $N = 100$ independent datasets in $\mathbb{R}^{n \times 2}$ with $n = 30$ total data points per dataset and $\kappa = 3$ distinct clusters. To generate the three clusters, we generate $n_1 = 5$ samples from $\mathcal{N}\left([1.5, 1.5]^\top, I_2\right)$, $n_2 = 5$ samples from $\mathcal{N}([3.5, 3.5]^\top, I_2)$, and $n_3 = 20$ samples from $\mathcal{N}([2.5, 2.5]^\top, 0.3^2 I_2)$, where $\mathcal{N}$ denotes a Gaussian distribution, and $I_2$ is a $2 \times 2$ identity matrix. An example of one of our generated datasets is shown in Figure 15.

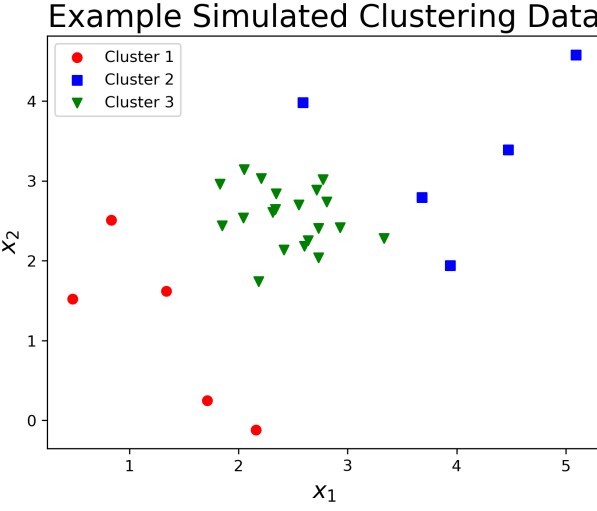

Figure 15: Example clustering dataset where the sample size is $n = 30$ and the number of clusters is $\kappa = 3$.

## G.2 Base algorithm $\mathcal{A}$: the "elbow" plot from $\kappa$-means

A common technique in unsupervised clustering for selecting the number of clusters $\kappa$ is via the "elbow" of the plot of the sum of squared distances (SSD) vs. $\kappa$. We deterministically select $\kappa$ as described in Algorithm 2, which selects the number of clusters by some hyperparameter for the $\kappa$ value that leads to the minimum observable slope of the SSD vs. $\kappa$ plot. In equation 26, $S_i$ is the set of points in cluster $i$, and $\mu_i$ is the cluster center for cluster $i$ selected via $\kappa$-means. In our generated example, the maximum number of clusters $M = 29$, and the slope tolerance $\omega = 5$. Additionally, for each run of $\kappa$-means, the clusters are randomly initialized.

---

**Algorithm 2** Computing the number of clusters

---

**input** Maximum number of clusters $M$, dataset $X$, slope tolerance $\omega$
   **for** $\kappa = 1, \ldots, M$ **do**
     Compute the sum of squared distances (SSD):

$$\text{SSD}_\kappa = \sum_{i=1}^{\kappa} \sum_{x_j \in S_i} ||x_j - \mu_i||^2, \tag{26}$$

     Compute the slope of SSD curve using the current $\text{SSD}_\kappa$ and the previous $\text{SSD}_{\kappa-1}$:

$$\text{slope}_\kappa = \frac{\text{SSD}_\kappa - \text{SSD}_{\kappa-1}}{\kappa - (\kappa - 1)}$$

     **if** $\text{slope}_\kappa < \omega$ **then**
        **output** Number of clusters $\kappa$
     **end if**
   **end for**
**output** Number of clusters $M$

---

## H Decision tree classification on single cell transcriptomics data

We investigate a model selection problem involving deciding the structure of a decision tree for multiclass classification. In this setting, we only measure two decision trees as distinct from each other if the trees split on the same features for a given split. Therefore, we do not consider the exact split value for continuous features in deciding whether two selected models are the same.

We explore the decision tree stability with data from single cell transcriptomics to classify stem cell fate based on gene expression. Mouse embryonic stem cells were sequenced for their gene expression and subsequently labeled as three different cell types: embryonic stem cells (ESC), epiblast-like intermediate cells (EPI), and neural progenitor cells (NPC) (Veleslavov and Stumpf, 2020). The cell fate label is noted to be assigned by clustering, so there is uncertainty in the true cell type label.

The dataset contains 547 cells and 96 gene expression values per cell. To explore stability in a setting where the sample size is much lower than the feature dimensionality, which is a common problem in genetics, we randomly subsample the number of cells to 50 and compute decision tree stabilities. As shown in Figure 16, gene expression is high correlation among genes, which may lead to instabilities in the fitted decision tree for small perturbations in the training data.

The base algorithm $\mathcal{A}$ corresponds to a multiclass classification tree. Broadly, decision trees create a sequence of splits on features that sort the training data samples into nodes, where a particular split is chosen to minimize the impurity of the samples in the two split nodes. Greedy algorithms are typically implemented to sequentially choose the locally best split at each node. More information on decision trees can be found in Breiman et al. (1984). Additionally, Xin et al. (2022) formulates an algorithm for finding the full *Rashomon set* for decision trees, which can be used as an alternative base algorithm to rank decision tree structures.

For this classification problem, we choose to measure dissimilarities among samples in a node via Gini impurity, which can be written as

$$\sum_{d=1}^{D} p_{ld}(1 - p_{ld}) \tag{27}$$

for $d \in [D]$ classes at node $l$, where the proportion of class $d$ in node $l$ is

$$p_{ld} = \frac{1}{n_l} \sum_{y \in Q_l} \mathbb{I}\{y = d\}, \tag{28}$$

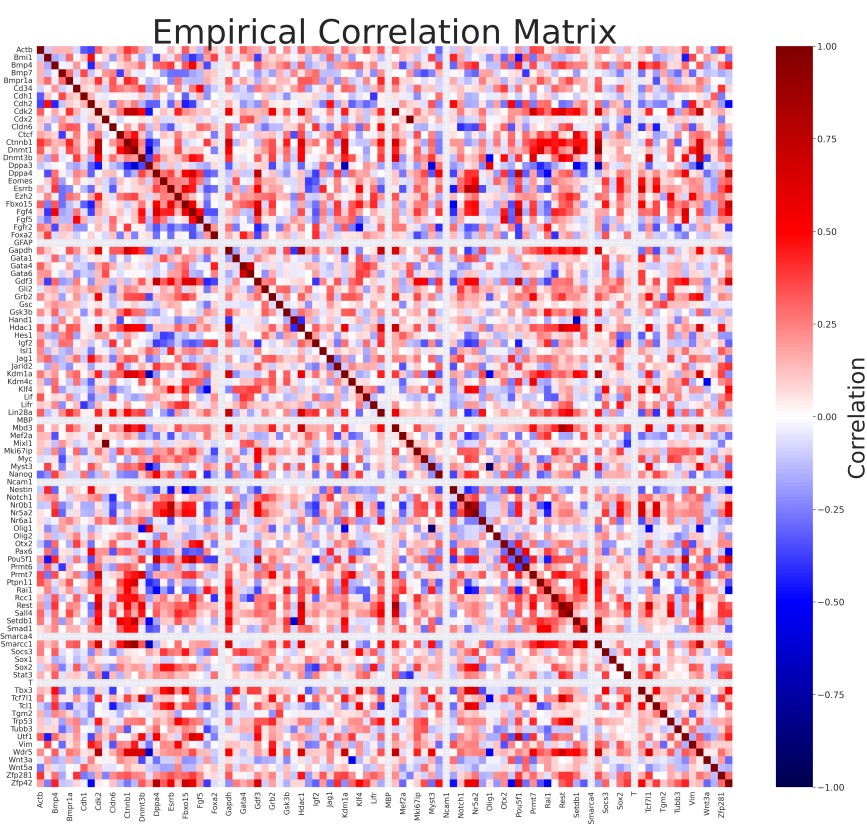

Figure 16: Empirical correlation matrix of gene expression across 50 cells from data in Veleslavov and Stumpf (2020). Rows and columns that show a null value correspond to genes that showed no variability in expression across the 50 cells.

and $Q_l$ denotes the split of node $l$ based on feature index $f$ and split value $t_l$:

$$Q_l^{\text{left}} = \{(x, y)|\ x_f \le t_l\}, \tag{29}$$

$$Q_l^{\text{right}} = Q_l \setminus Q_l^{\text{left}}. \tag{30}$$

Our experiments use the default hyper-parameters in the `sklearn.tree.DecisionTreeClassifier` function.

### H.1 Results

Since we have one dataset of gene expression across cells, $N = 1$ in this example. Table 4 shows the stability results and average number of models returned across LOO trials. Notably, $\text{ip}_\tau$ is absent from this table because this selection method is specific to variable selection, which is a special class of model selection.

Based on Table 4, significant instabilities are seen for both the $\text{argmax} \circ \mathcal{A}$ and $\text{argmax} \circ \tilde{\mathcal{A}}_{K,B}$ model selection methods, while top-2 $\circ \tilde{\mathcal{A}}_{K,B}$ provides enhanced stability. For the inflated argmax, we chose $\varepsilon = 0.03$ so that the stability matched that of the stability of top-2 in order for us to directly compare the number of returned models. For a fixed level of instability $\delta = 0.12$, the inflated argmax returns a smaller number of models on average across LOO trials, showing that our method adaptively returns the smallest set of models based on the underlying uncertainty in the selection.

For $\varepsilon = 0.04$, we obtain 0.04 instability, corresponding to a high degree of stability with a LOO average of 2.88 returned models. We visualize the top 3 selected models across bootstrap samples in Figure 17. From these trees, we can see that the first and second most common selected trees across bootstrap samples are completely disjoint, potentially indicating strong correlations among the features for both trees. However, the third most commonly selected tree differs from the top tree by only one split, suggesting there may be a high correlation between the gene expressions of Lin28a and Nanog. Future investigations can focus on disentangling the associations among the genes selected in the top tree versus the second most selected tree to elucidate their associations with the cell fates.

| SELECTION METHOD | LOO INSTABILITY | AVG. LOO SET SIZE |
|---|---|---|
| $\text{argmax} \circ \mathcal{A}$ | 0.40 | 1.00 |
| $\text{argmax} \circ \tilde{\mathcal{A}}_{K,B}$ | 0.24 | 1.00 |
| top-2 $\circ \tilde{\mathcal{A}}_{K,B}$ | 0.12 | 2.02 |
| (Ours) $\text{argmax}^{0.03} \circ \tilde{\mathcal{A}}_{K,B}$ | 0.12 | 1.74 |

Table 4: Table of stability and the average number of models returned across LOO trials for selecting the structure of the multi-class decision tree for a gene expression dataset from Veleslavov and Stumpf (2020). In this experiment, $\mathcal{A}$ is a classification decision tree, and $\tilde{\mathcal{A}}_{K,B}$ is a bootstrapped classification decision tree with $B = 10,000$ and $K = 45$. In computing the returned models, if there is an exact tie in the top models, all ties are returned, which explains why top-2 returns slightly more than 2 models on average across the LOO fits.

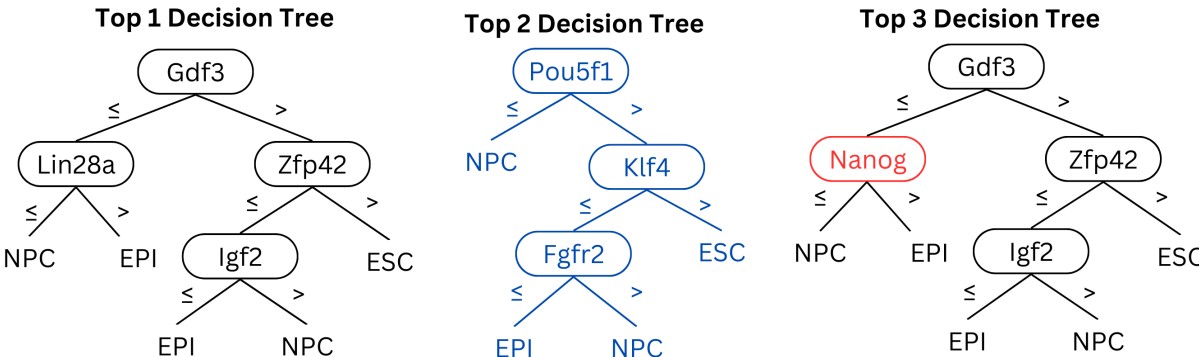

Figure 17: Visualization of the decision tree structures for the top 2 trees chosen by bootstrap sampling the data for $K = 45$ samples per bag across $B = 10,000$ bags. Differences between the top and second tree are indicated in blue in the second tree, and the differences between the first and third trees are indicated in red. The names of the genes that determine the split at each node are shown in ovals, and the class labels NPC, ESC, and EPI are listed for each leaf. The top 1 decision tree was selected by 4.74% of bags, the top 2 decision tree was selected by 2.80% of bags, and the third decision tree was selected by 2.24% of bags. The exact values used for the splits at each node were not considered in our stability results.

