# OpenReview forum: "Stabilizing black-box model selection with the inflated argmax"
_TMLR — Accepted by TMLR_

### Review · Reviewer_X5xN · 2025-08-04

**Summary Of Contributions:**

# Summary

The authors consider the problem of stable model selection, i.e., model selection which gives consistent results even in the presence of small changes to the dataset. They define a novel notion of model selection stability which applies to methods which return a set of possible models, given a dataset. The notion measures the frequency with which the set of models selected by applying the selection algorithm to the entire dataset is disjoint from the set of models selected when one point is removed from the dataset. Inspired by methods from stable classification, they propose the inflated argmax for stable model selection. They prove that models selected by the inflated argmax are guaranteed to be stable with respect to their proposed notion of model stability. Finally, they validate their method with three experiments, two on synthetic data and one on real flow cytometry data.

# Strengths

The fundamental problem considered (stable model selection) is interesting and useful. The shortcomings of existing selection procedures (e.g., when several models are near optimal, selecting the argmax will be unstable; if the selection probability of several models is close to the selection threshold, this will also be unstable) were clearly explained. Framing the problem of model selection in a multiclass classification framework in order to employ techniques from that literature is also an interesting connection.

# Weaknesses

For major weaknesses, see the discussions below.

Typos/notational clarifications:
- In Definition 4.1, what is the norm being used in $||q-w||$? Is it the L1 norm of the probability distributions considered as vectors?
- What is the value of $\varepsilon$ in Fig. 1?
- At the end of Section 5.1.1 it is stated that the base algorithm $\mathcal{A}$ for the experiment is SINDy, while then end of Section 5.1.2 it is stated that the problem is solved using STRIdge penalization as the base algorithm.
- The formatting of the headings of Table 2 should be fixed. There is no spacing between the headings, making it difficult to read.

**Additional Comments:**

There are some other recent works on model set selection which would be interesting to discuss. Specifically, there has been some recent interest in constructing Rashomon sets of models, which are defined as model sets which have performance close to that of the single best model on a given dataset. See, e.g.:

>Xin, R., Zhong, C., Chen, Z., Takagi, T., Seltzer, M., & Rudin, C. (2022). Exploring the whole rashomon set of sparse decision trees. Advances in neural information processing systems, 35, 14071-14084.

This particular work focuses on sparse decision trees in particular, so the methods proposed in the submitted manuscript are more general. Nevertheless, it seems like there would be some strong connections between model selection stability considered in the submitted manuscript and sets of near-optimal models considered by these previous works. This connection should at least be mentioned and I believe it would strengthen the paper to discuss the connection (or dissimilarity) in some detail.

**Audience:**

Yes

**Audience Explanation:**

The problem of stable model selection is of interest to TMLR's audience, but I don't believe the paper has adequately motivated why the notion of stability that it proposes is a reasonable one. In particular, the definition of model selection stability (Definition 2.1) seems unfairly biased against simple model selection procedures. For instance, if the model selection procedure selects $\hat{M}$ which contains 1000 models, and after removing a single datapoint, the resulting $\hat{M}^{\setminus i}$ differs from the original in 999 of the 1000 models but has a single model as overlap, this will not be penalized at all in the stability calculation, even though intuitively it seems that this is a very unstable selection procedure. A natural fix to this is to use the IoU of the two selection sets, i.e.,

$$\frac{1}{n}\sum_{i=1}^n \left(1 - \frac{|\hat{M}\cap \hat{M}^{\setminus i}|}{|\hat{M} \cup \hat{M}^{\setminus i}|}\right).$$

When $\mathcal{M}$ is a simple model selection procedure, this reduces to the formula in equation (2). In the case of the example above, such an example would still be highly penalized (it would add 1998/1999 to the sum).

In order to meet the TMLR criteria, I believe some justification should be provided for why the proposed stability notion is still reasonable in light of this example.

---
Post rebuttal: In light of the discussion and the other reviewers' interest, I have updated my score. I still believe a more thorough discussion of the limitations of the stability notion (implicitly favoring larger decision sets; the need for also considering utility) should be included to give readers a complete picture.

**Claims And Evidence:**

Yes

**Claims Explanation:**

The evaluation criteria used in the experiments are insufficient for showing the benefit of the proposed method, specifically on the real data.

In particular, considering only stability without a notion of correctness is not convincing, because any algorithm which outputs a single model regardless of the input dataset is perfectly stable while also outputting a prediction set of minimal size. So in the real dataset, such an algorithm would have obtained the best results if it was only evaluated according to the metrics reported in Table 2.

For the Lotka-Volterra model experiment, the results include a notion of correctness (utility-weighted accuracy, equation (7)) which also takes into account the number of models returned by a given method. This is an improvement over the real data experiment. Unfortunately, the results are not favorable for the proposed method. For the two strictest worst-case instability levels, the inflated argmax obtains 0% utility weighted accuracy (Fig. 3, middle plot). This seems especially problematic as it is stated that "our primary interest is in small $\delta$ (i.e., high stability)" (pg. 7, after equation (6)).

---
Post rebuttal: The proposed method produces several model sets on the Pareto frontier in the Lotka-Volterra experiment. The discussion of utility-weighted accuracy was also helpful. In order for the flow cytometry results to be meaningful, however, some notion of correctness (even a qualitative discussion based on domain knowledge) is necessary.

**Requested Changes:**

1. Please explain how the proposed stability notion can be considered useful in light of the example provided above.

2. Please provide some objective measures of correctness on the real data experiment.

3. How should the results showing 0% accuracy at high levels of stability (small $\delta$) in Fig. 3 be interpreted?

---

> ### Author Response · Authors · 2025-09-18
>
> Thank you for your review of our work. We have uploaded a revised version of our paper, and below is our response to your questions and comments.
>
> "In Definition 4.1 what is the norm [...]" It is the Euclidean norm of the distance between two probability vectors $w$ and $q$. We have updated the paper to state this.
>
> "What is the value of $\varepsilon$ in Fig. 1?" Figure 1 serves as an illustration of our method to emphasize that the inflated argmax allows us to return multiple models if there is a high degree of uncertainty, such as when the weight vector $\hat w$ lands in the overlapping regions defined by $\varepsilon$. Since the figure is an illustrative example, there is no particularly chosen value associated with $\varepsilon$. However, based on how wide the region defined by $\varepsilon$ spans in the illustration, we would estimate it to be $0.2$.
>
> "At the end of Section 5.1.1 [...]" SINDy is a general approach for formulating a problem to identify the governing equations for a dynamical system in terms of penalized regression. This framework can be specified with a variety of different penalization choices and algorithms, and one particular choice is STRidge penalization. In the updated version of the paper, we clarified that SINDy is a generic problem formulation for dynamical system equation discovery, and STRidge is the particular algorithm we used to solve this problem formulation.
>
> "The formatting of the headings of Table 2 [...]" Thank you for pointing this out. We have edited the table to improve readability.
>
> "Please explain how the proposed stability notion can be considered useful in light of the example provided above." We thank the reviewer for the interesting proposed reformulation of our defined stability measure. What we propose is perhaps the mildest possible stability criterion. We just require some overlap in the returned selected models when one sample is removed. Surprisingly, prior to our submission, there were no theoretical guarantees of even this mild criterion, and as demonstrated in our empirical results, even this mild stability criterion is frequently violated by commonly-used methods. Given this observation, it is clear that our manuscript makes a substantial contribution to the literature. The reviewer's proposed metric is a stronger notion of stability, so a method that does not have stability by our metric will also not be stable by their metric. Proving stability guarantees for an entirely new metric would be interesting in future work, but our work is a necessary first step towards methods with any stability guarantees.
>
> "Please provide some objective measures of correctness on the real data experiment." To our understanding, a ''ground truth'' model for this problem is not fully known, though some connections within the graph have empirical evidence based on domain knowledge and experimental data. Due to the uncertainty in the true model, we cannot provide a measure of correctness like we could in our simulated experiments. We emphasize that our method is guaranteed to retain stability, which we convincingly show for this problem in Table 2. For the simulated experimental results shown in Figure 3, we show that we achieve this stability non-trivially; stability is not achieved by sacrificing accuracy of returning the correct model (See our reply to "How should the results showing 0% accuracy [...]" for more explanation).

---

> > ### Author Response · Authors · 2025-09-18
> >
> > "How should the results showing 0% accuracy at high levels of stability (small $\delta$) in Fig. 3 be interpreted?" We emphasize that Fig. 3 plots the utility-weighted accuracy, not just accuracy. The utility-weighted accuracy down-weights the accuracy measure depending on the number of models returned. For example, if $|\hat{M}|= 1$, and the true model is contained in $\hat M$ (i.e., $m^\* \in \hat{M})$, then both the accuracy and the utility-weighted accuracy are 100\%. However, if $|\hat M|=1,000$ and $m^* \in \hat M$, the accuracy is 100%, but the utility-weighted accuracy is $0.1%$. The utility-weighted accuracy measure aims to identify returned model sets that are both accurate and interpretable, and we argue that a smaller number of returned models is more interpretable. In Figure 3, our method provides near 0% utility weighted accuracy as $\varepsilon$ increases because an increasing $\varepsilon$ results in larger returned model sets $\hat M$, leading to a decrease in the utility-weighted accuracy, even though the correct model $m^*$ is contained in this set.
> >
> > The reviewer is correct that our interest is in high stability (low $\delta$) regimes. Our framework results in a tradeoff between stability and the number of models returned via $(\varepsilon,\delta)$ pairs, and this tradeoff can be navigated by the user. With our framework, increased stability may be achieved by returning a larger set of models in $M^+$. The user's stance on this tradeoff can be expressed through the choice of $\varepsilon$, which directly relates to the user's worst-case instability tolerance through Theorem 4.2. Our empirical results show that there are $(\varepsilon,\delta)$ pairs that give high utility-weighted accuracy and low instability for many problems, resulting in returned sets that are both accurate in containing the ''true'' model and interpretable in that the returned sets are as small as possible.
> >
> > "There are some other recent works on model set selection [...] (e.g., Xin et al. 2022) [...]" Thank you for bringing the Xin et al. 2022 paper to our attention, as it is a relevant work that we have cited in an updated version of the paper. As pointed out, this particular work is interested in decision trees, which is a particular model selection setting, whereas our approach applies to any model selection procedure. Moreover, our work is concerned with the stability of the returned set, and we provide theoretical guarantees for this stability, whereas Xin et al. 2022 is primarily concerned with building an exact algorithm for constructing the full Rashomon set for decision trees. While it is possible that a Rashomon set may be stable in some settings, there is no  theoretical stability guarantee similar to what we provide for our returned sets of models. Xin et al. 2022 provides an example (Figure 5 in Xin et al. 2022) that illustrates the differences in the fitted tree structure when training on the full dataset compared to removing 1\% of the data. In our paper, we more rigorously explore this comparison and define stability with a milder perturbation to the data: a perturbation where only one data point is removed from the training dataset. Since this paper explores similar ideas in the context of decision trees, we have cited this paper in the updated version to refer the reader to it as a relevant work in the case of decision trees. We have additionally connected the similarity of our approach to the idea of Rashomon sets/the Rashomon effect presented in Leo Beiman's 2001 paper and have cited it in the related work section.

---

> > > ### Comment · Reviewer_X5xN · 2025-10-04
> > >
> > > Thanks to the authors for their thorough response. I have also looked at the other reviews.
> > >
> > > ## Interest criterion
> > >
> > > In light of the fact that there are no existing theoretical results for the proposed mild stability notion, and the fact that it can be violated in practice, I agree with the authors that their result makes a meaningful contribution. The other reviewers have also expressed interest in the proposed notion. I have changed my evaluation accordingly. Nevertheless, I believe the paper should include a more thorough discussion of limitations, specifically, the inherent advantage the proposed notion gives to large model sets. The last line of the conclusion is a step in the right direction, but I believe that reference to a concrete example would give a more complete picture to readers.
> > >
> > > ## Evidence criterion
> > >
> > > In light of the discussion of utility weighted accuracy and the fact that the proposed method produces many model sets along the Pareto fronteir in the Lotka-Volterra model, I have also updated my evaluation for this criterion. I appreciate that the ground truth for the flow cytometry experiment is not fully known, but in light of the previous discussion, I still believe *some* measure of correctness must be given for the results to be meaningful. Even a qualitative discussion of the connections in the graph which align with domain expertise would be helpful, since the fact remains that as is, without further discussion, a constant algorithm which returns a single graph would appear to have the best performance according to the criteria in the paper.

---

> > > > ### Author Response · Authors · 2025-10-22
> > > >
> > > > We greatly appreciate your consideration of our work and our replies. We have uploaded a new version of the paper with a few additions that aim to address your most recent points.
> > > >
> > > > To address the concern brought up in the "Interest criterion" section of your recent reply, we have restructured and added to our discussion section. We especially point the reviewer to the part of the discussion section that states: "We believe another interesting direction is developing similar theoretical stability results for other measures of stability aside from Definition 2.1. Our definition of stability is relatively weak, reflecting a bare minimum criterion necessary (though not sufficient) for many other notions of stability. [...]."
> > > >
> > > > Additionally, we appreciate the consideration for some basis of ground truth in the graph experiment noted in the "Evidence criterion" section of your reply, so we have added the last paragraph in Section 5.2.1.: "Though there is no agreed-upon 'ground truth' graph connection structure [...]."

---

### Review · Reviewer_TSM4 · 2025-08-12

**Summary Of Contributions:**

The paper is motivated by the observation that standard frequentist model selection procedures can be very unstable, potentially returning different results even when a single observation is dropped. The authors propose to stabilize black-box model selection algorithms by (1) using bagging to obtain weights for each model and (2) applying an "inflated argmax" procedure to select a set of top models that is stable across dropping observations. The authors show that this procedure provides to a precise form of "model selection stability." Their theoretical result relates the degree of stability to algorithm's parameters, which aids users in selecting these parameters. Experimental result show that the proposed approach provides equal or greater stability compared to alternatives.

Strengths:

+ easy-to-use, black-box method
+ works well empirically
+ simple but precise theory that guides algorithm tuning

Weaknesses:

- leave-one-out stability is a fairly weak guarantee
- unclear why the goal of selecting a stable "top" model is practically relevant

**Audience:**

Yes

**Audience Explanation:**

Black box model selection is an important and common problem in ML. Users of ML methods want their procedures to be "robust," including being stable to small changes in the data. So, I could see this method being further developed and/or used in practice.

**Claims And Evidence:**

Yes

**Claims Explanation:**

**Claim 1:** *Our method selects a small collection of models that all fit the data*

Supported by experimental results – 1 simulation study, 2 real data examples, all using quite different models

**Claim 2:** *...and it is stable in that, with high probability, the removal of any training point will result in a collection of selected models that overlaps with the original collection.*

Supported Theorem 4.2

**Claim 3:** * the proposed method yields stable, compact, and accurate collections of selected models, outperforming a variety of benchmarks.*

Supported by experimental results.

**Requested Changes:**

I have two requested changes, both of which aim at clarifying the motivation and better justifying the specific goals of the work:

1. **Justification of stability criterion:**  The leave-one-out stability criterion seems quite weak. For example, the theory from Huggins \& Miller (2023) shows that BayesBag can be stable even when the full dataset is resampled. So, I think this choice needs better justification.

2. **Goal justification:** I would like to see some justification for why choosing a single (or small set) of "top" models is a sensible goal. Indeed, once we effectively have a distribution over models, why not select the smallest set of models with total probability at least $1 - \delta$? This seems like a more interpretable goal, as it aims to guarantee the "true" top model is getting returned. For example, if the models have weights (.35, .35, .25, .1), the inflated argmax might select models 1 and 2 but leave out model 3, even though the third model still has considerable weight. The 90% "credible set," on the other hand, would return models 1–3.

---

> ### Author Response · Authors · 2025-09-18
>
> Thank you for your review of our work. We have uploaded a revised version of our paper, and below is our response to your questions and comments.
>
> "The leave-one-out stability criterion seems quite weak. For example, the theory from Huggins and Miller (2023) shows that BayesBag can be stable even when the full dataset is resampled. So, I think this choice needs better justification." Please see our reply to Reviewer X5xN under the question "Please explain how the proposed stability notion can be considered useful in light of the example provided above."
>
> Additionally, BayesBag is interested in stability of the estimated posterior distribution. A higher degree of stability is achieved by averaging posterior model probabilities computed across multiple bootstrapped datasets, thereby smoothing out changes in the estimated posterior distribution with perturbations to the underlying data. The notion of stability that we consider in this work is not in the estimates of the model weights, however, it is in the choice of final selected model. We mention that Soloff et al. (2024a;c) shows that bagged weights are stable (i.e., $\hat w$ does not change much in the Euclidean norm with perturbations to the underlying data); however, our results show that even with this approach, a bagged base algorithm (denoted as $\tilde{\mathcal{A}}_{K,B}$) is still unstable in terms of the final model selected when applying the argmax as a selection criterion for returning the best model from removing just one datapoint. One can expect sometimes even larger instability if more datapoints are removed at once (or the entire dataset resampled as the reviewer mentions).  Using our method provides enhanced stability with theoretical guarantees; similar theoretical guarantees are not present in Huggins & Miller (2023).
>
> "I would like to see some justification for why choosing a single (or small set) of "top" models is a sensible goal. [...]" The reviewer's suggested goal of returning models that have estimated probability at least $1-\delta$ for some choice of $\delta \in [0,1]$ is a useful goal, however, our problem aims to address a different target. What the reviewer calls the ''credible set'' perhaps may also exhibit arbitrary cutoffs, where only some models with very similar probabilities are included by virtue of contributing to the sum to 0.9, despite there being other models with nearly similar estimated probability that are not included. Consider an example where $\hat w = (0.45, 0.35, 0.11, 0.09)$. If we return a 90\% credible set, this would return $(m_1, m_2, m_3)$ since $0.45+0.35+0.11 \geq0.9$. However, the difference in predicted probabilities for models $m_3$ and $m_4$ is minimal, and small perturbations to the training data could lead to these models swapping. The inflated argmax, however, may return $(m_1,m_2,m_3,m_4)$ since the estimates probabilities for $m_3$ and $m_4$ are nearly indistinguishable. Therefore, the inflated argmax is more likely to return sets of similarly performing models. In addition, our method does not require the bagged weights to correspond to model posterior probabilities, which would be necessary to interpret the outcome of the reviewer’s approach as a credible set.
>
> Theoretical stability guarantees of $\hat M$ to our knowledge have not been established for this ''credible set'' approach, which could be a potentially interesting direction for future work to investigate. Note, however, that our framework is guaranteed to provide model selection stability even if the weights cannot be interpreted formally as Bayesian posterior probabilities.

---

### Review · Reviewer_TwV9 · 2025-09-01

**Summary Of Contributions:**

In this work, the authors introduce a new technique for model selection. The paper is clear and well written. I found the proposed technique new and the multi-class classification perspective is fresh and interesting. From there, using the newly introduced inflated argmax is an interesting choice. The technique only requires one interpretable hyperparameter. The experimental results provide reasonable evidence that the technique has practical merits.

**Audience:**

Yes

**Audience Explanation:**

Although there is vast literature on the topic of model selection, this paper brings a new perspective. I think that TMLR readers and practitioners will find this paper interesting.

**Broader Impact Concerns:**

Model selection, although of critical importance in machine learning, can be clearly misused to infer all sorts of misleading results if applied without care. As such, the authors should probably add a paragraph explaining that a model that exhibits high stability does not necessarily imply that the "ground truth" model was found with high probability or anything remotely similar. Of course, this does not subtract value form the contribution, but such a general statement would be highly welcomed.

**Claims And Evidence:**

Yes

**Claims Explanation:**

The new perspective proposed in the paper is well justified and supported by experimental results. The idea is intuitive enough that its introduction and discussion would be of interested to the community.

**Requested Changes:**

I have a few questions/comments (in no particular order):

A- In section 2, model selection is described as choosing "the model(s) [...] that provide the best fit to the data." I believe that this description is slightly misleading as fitting is not the only quantity involved in model selection. For example, in variable selection, selecting all variables will (in general) fit the data better than selecting a subset but this complete model is often not desirable. It is clear to me that the authors understand this difference, but a discussion around this topic might help practitioners reading the paper and trying to use their technique.

B- Although one of the claims in the paper is that the proposed approach extends beyond variable selection, all experiments in the main manuscript only involve variable selection. There is one experiment involving decision trees in Appendix F (not mentioned in the main text). I would have appreciated one example (and preferably two) in the main text to support this claim. As a suggestion, selecting k in k-means is a classical example that the authors mention in the introduction. There are many techniques/measures for this algorithm (the silhouette score, the adjusted Rand index, adjusted mutual information, and information-theoretic approaches, for example). Although I understand that algorithmic-specific techniques may yield better results than a general technique like the one proposed here, it would be interesting to see the differences. Moreover, could these standard measures be used as algorithm A in the proposed framework?

C- The clustering literature has analyzed the problem of the relationship between the selected number of clusters and the algorithmic stability of the optimization techniques under subsampling. See [Shai Ben-David, Dávid Pál & Hans Ulrich Simon. Stability of k-Means Clustering. COLT 2007], for example, where a negative association between both concepts if formally proven. The authors should discuss how these results relate to their approach.

D- There is also ample theory on model selection for the LASSO itself. Although I understand that LASSO can be regarded as a form of variable selection technique, it is a model in itself. It commonly comes with a hyperparameter controlling the strength of the L1 regularization. In the paper, these type of parameters are described as "optimization hyperparameters" (in Appendix D.1.2, for example). However, their values affect directly the cardinality of the solution in non-trivial ways. Hence, their values cannot be optimized independently, especially in the conditions analyzed in the paper (correlated variables) as the ensuing choice will be (as recognized by the authors) unstable. Would it be possible to discretize the range of this parameter and fold it into the proposed framework? Evaluating this experiment would be very interesting.

E- I understand the statistical merits of returning multiple models instead of just one when this selection is unstable. However, I would like to see a discussion of what it means for the practitioner. How to deal with multiple models? In certain areas, like decision trees, one can just treat them as a forest, for example. In other cases, the interpretation is more uncertain. For variable selection, what would happen if the variables selected by two returned models do not overlap (or overlap only slightly)?

F- The main stability result links the fraction of samples to the stability to the selection. Has the set of "instability inducing" samples any interpretable meaning? Are these samples just outliers? What should the practitioner do with these samples (if anything at all)?

G- What happens if there are no models at the required confidence level? Can this even occur? Would lowering the confidence level just solve the issue? For problems where the model category (e.g., LASSO) is a crude approximation of the underlying data is delta stable? In the sense that the proposed technique should not transition sharply from returning no models to returning "all" models as delta is relaxed.

H- The proposed technique comes with a stability guarantee but Theorem 4.2 has no proof. If its proof follows immediately that of Theorem 17 by Soloff et al., please state this in the manuscript. If it does not follow immediately, a proof is needed. Without this clarification, the theory feels vacuous.

I- What about uncountable set of models (with a continuous parameter, for example)? If this is out of scope, the authors should clearly identify this problem constraint where they state the main claims of the paper (abstract, Section 1.1, conclusions). If it is not out of scope, they should describe how to deal with such a scenario.

J- The statement in Section 6 "We define as notion of stability [...] that computes the proportion of LOO selected models that are disjoint from the selection made from by training on the full data" seems slightly misleading. Since bagging is used, no model is trained on the full data. Did I miss a critical step or should the wording be adjusted?

Minor points:
- Appendices D, E, and F are not mentioned in the main text.
- There is extensive literature pertaining model selection. The overview of the topic in the paper seems to be constraint to a specific type of techniques (i.e., bagging). I believe that the authors should devote a couple of sentences to the bigger picture.
- Paragraph after Equation (1) in the sentence "this weight can be thought of as a probability of each candidate model," the authors should clarify which event this probability is related to.
- In Section 2.1, I found the sentence "We denote bagging a base algorithm A as A_{K,B}" very unclear. Maybe something like "We denote by A_{K,B} the algorithm resulting from applying a bagging technique on algorithm A" would be clearer. There may be better formulations that the one I propose here.
- In section 2.1, the sentence "the ensemble weight w_m simply counts the fraction of bags wherein model m was selected" is unclear and needs more elaboration. Maybe clarify that for simple model selection each vector w^b is one-hot would help.
- In section 2.1, "A_{K,B} provides a more stability" -> "A_{K,B} provides more stability".
At the start of Section 2.2, the notation "\hat{M} \in \argmax_m \hat{w}_m" implies that \hat{M} is a single element and not a set. Replacing "\in" by "\subseteq" may be a better choice.
- I believe that should Definition 2.1 involve "a given dataset D" instead of "all datasets"? The measure s dataset specific.
- In Section 4.1, "denotd" -> "denote".
- In Definition 4.1, which norm is used in "dist"?

---

> ### Author Response · Authors · 2025-09-18
>
> Thank you for your careful review of our work. We have uploaded a revised version of our paper, and below is our response to your questions and comments.
>
> A - We appreciate the nuanced point that the reviewer brings up regarding the  definition we present for model selection. We agree that the wording was perhaps misleading, so we have reframed the definition to state "the model(s) [...] that provide the best performance as a function of the training data, where performance may reflect some combination of fit to data and structural properties of the model." This definition is a bit more broad since performance may be measured in a number of ways.
>
> B - Thank you for your suggestion that has made the main text of our paper stronger. We have included a new k-means clustering example in the main text and have referenced our decision tree example in the appendix as another example of a model selection example that does not involve variable selection. We have chosen the base algorithm $\mathcal{A}$ in the clustering example to be an algorithm that selects the number of clusters in the data based on the sum of squared distances (SSD) via k-means clustering across various choices of the number of clusters. Details are shown in Section 5.3 in the main text and Appendix G for this new example.
>
> One could easily consider alternative base clustering algorithms within our framework, e.g., based on the silhouette score, etc. However, the point of our paper is not to compare how our method performs empirically across different base algorithms with different objectives, but rather to present a general framework with theoretical stability guarantees, and we observe that it is efficacious across various empirical examples.
>
> C - Thank you for pointing us to this interesting work. We note that the notion of stability presented in Ben-David et al. 2007 defines it in terms of the distance of the expected cluster centers across many subsampled datasets asymptotically as the data sample size approaches infinity. Our paper would define stability in the example of k-means as the stability of the choice of $k$, which does not consider the stability of the fitted clusters themselves. Additionally, the theoretical guarantee we present in our work applies to finite-sample datasets, which provides a more practical theoretical guarantee than theory based on sample size asymptotics. The issue of stability of the cluster centers is an interesting notion to theoretically analyze, but our approach aims to address a related but slightly different problem.
>
> D - While the reviewer is correct in that the choice of the parameters we call ''optimization hyperparameters'' nontrivially impacts the cardinality of the solution, we emphasize that our method will yield stable model selection even if the user chose hyperparameters poorly. The fact that our approach does not rely on specific features of the base algorithm makes it much more general purpose than a LASSO-specific stability guarantee. We agree with the reviewer that a potentially interesting direction would be to jointly consider accuracy and stability. Since our main contribution lies in stability, however, we will leave this direction to future work and have mentioned it as a potentially interesting future direction of our work in the discussion section.
>
> E - The issue of how to deal with multiple models returned is certainly problem-specific, however, we included in the discussion section a brief discussion of options for practitioners. We added the following paragraph to our paper "A natural follow up question to our approach is what a practitioner should do if multiple models are returned. As is the case in many practical settings, the answer to this question is problem-dependent. As noted in Section 5.2.1, top models that substantially overlap can prompt directed follow-up experiments. Moreover, the size of the set of returned models may be a useful indicator of model uncertainty for a practitioner. The decision for how to handle multiple models should be carefully considered and align with the practitioner's ultimate goals (e.g., model interpretation or predictive accuracy)."
>
> F - The set of samples that, when removed from the training data for model selection, induce instability are not necessarily outliers for a meaningful reason. In all our simulated experiments, all data points were generated from the same underlying distribution, and we still saw substantial instability in the model selection. A particular point inducing instability does not necessarily make it an outlier or an interesting point; therefore, there is not necessarily a meaningful interpretation for these points.

---

> > ### Author Response · Authors · 2025-09-19
> >
> > G - Our approach is guaranteed to return at least one model. The most likely model given by the weights $\hat w$ will always be returned. Aside from the top model, the parameter $\varepsilon$ controls whether any other models are returned in addition. For example, if the weight vector is $\hat w = [0.51,0.49]$ and $\varepsilon = 0.02$, then both models will be returned since perturbing the weights with a magnitude at most 0.02 can result in the vector $[0.49, 0.51]$, in which case, the second model is then the most likely model. If $\varepsilon=0$, which is equivalent to the argmax operation, only model 1 will be returned. It can certainly be the case that $\mathcal{A}$ is very uncertain about which model is the best model, leading to entries of $\hat w$ with very low values. Even in this case, at least the top model will be returned.
> >
> > In terms of the potential for the inflated argmax to sharply return "all" models, we agree that with our approach, a graceful increasing in the number of models returned for decreasing $\delta$ is not necessarily guaranteed. Indeed, if the bagged model weights were all equal except for the best model, which had only slightly higher weight, then our method would either return only the best model or all the models. We empirically see that as $\varepsilon$ becomes larger (and therefore the worst-case instability $\delta$ becomes smaller), the number of returned models can sharply increase based on our experiments (right-most plots in Figures 3, 6, and 8). The degree to which this sharpness occurs depends on the non-uniformity of the ordered weights in $\hat w$ for the particular problem.
> >
> > H - We have edited the manuscript to note that the proof follows immediately from Theorem 17 in Soloff et al.
> >
> > I - We note before equation (1) that the set of all candidate models $M^+$ is countable.
> >
> > J - We appreciate the reviewer pointing out this subtlety as it may impact the understanding for other readers as well. The distinction is that "training on the full data" means that the full dataset is used to draw samples for each bag, whereas training on the LOO data means that the full data without one particular training sample is used to draw samples for each bag. The reviewer is correct that it is very possible that all samples do not appear across any of the bag despite the full dataset being accessible while constructing the bags. We have clarified this subtle point in the updated version of the paper by changing "training on the full data" to "training with access to the full data."
> >
> > "There is extensive literature pertaining model selection. [...]" The main contribution of our paper is the stability guarantee that we demonstrate for model selection problems. The emphasis of our paper is therefore not necessarily the algorithms used to solve particular problems (e.g., there are many other sparse linear regression algorithms other than LASSO), but rather the approach for stabilizing model selection procedures. Therefore, our related work focuses on bagging, which is a common approach in the literature that aims to address instability. We point the reader towards these papers for more context as to previous attempts in the literature to address the issue of stability and define in other contexts, including classification and estimation of coefficients in linear models. We have edited the related work section to include a few additional references that speak to other aspects of model selection that are broadly of interest. We additionally make the point clearer that bagging has historically been used to address concerns of stability.
> >
> > "Paragraph after Equation (1) in the sentence ``this weight can be thought of as a probability of each candidate model," the authors should clarify which event this probability is related to." We have edited the main text to
> > ''This weight may be loosely interpreted as the probability of the candidate model being the best model given the data among the choices in $M^+$.”
> >
> > "I believe that should Definition 2.1 involve 'a given dataset D' instead of 'all datasets'? The measure s dataset specific." The main result of this theorem is that the proof is not based on any particular properties of the dataset other than its sample size, so the theoretical guarantee applies for all datasets of that sample size.
> >
> > "The authors should probably add a paragraph explaining that a model that exhibits high stability does not necessarily imply that the ''ground truth" model was found with high probability or anything remotely similar." We included the following sentence  as a footnote while introducing model selection: ``Note that this is a general formulation of the model selection problem, and does not assume that there is a true data generating model in $M^+$."
> >
> > We thank you for noting the typos or areas where the language is unclear. We have updated the language accordingly in the revised paper.

---

### Decision · Action_Editor_ZNfh · 2025-10-29

**Recommendation:** Accept as is

**Audience:**

Yes

**Audience Explanation:**

The paper provides a new perspective on model selection and I agree with the reviewers that the proposed framework is likely to inspire additional follow-up work.

**Claims And Evidence:**

Yes

**Claims Explanation:**

The paper introduces a framework for stabilizing model selection and illustrates its broad efficacy on a range of very different synthetic and real world applications. Although the ground truth is not known in some of the real-world applications (i.e., the flow cyclometry example), the authors have provided evidence that the identified graphical models are scientifically plausible.